# Controls on the grain size distribution of landslides in Taiwan: the influence of drop height, scar depth and bedrock strength.

Odin Marc[1, 2, 3], Jens M. Turowski[2], and Patrick Meunier[4]

[1]Géosciences Environnement Toulouse (GET), UMR 5563, CNRS/IRD/CNES/UPS, Observatoire Midi-Pyrenees, 14 Avenue Edouard Belin, 31400 Toulouse, France.
[2]German Research Center for Geoscience, GFZ-Potsdam, Section 4.6, Geomorphology, Telegrafenberg, Potsdam, Germany
[3]Laboratoire Cogitamus, France, https://www.cogitamus.fr/
[4]Laboratoire de Géologie, Ecole Normale Supérieure, 15 Rue Lhomond, Paris, France

**Correspondence:** Odin Marc, odin.marc@get.omp.eu

**Abstract.** The size of grains delivered to rivers by hillslope processes is thought to be a key factor controlling sediment transport, and long-term erosion as well as the information recorded in sedimentary archives. Recently, models have been developed for estimating the grain size distribution produced in soil, but they may not apply to active orogens where high erosion rates on hillslopes are driven by landsliding. Until now, relatively few studies have focused on landslide grain size

distributions. Here we present grain size distributions (GSDs) obtained by grid-by-number sampling on 17 recent landslide deposits in Taiwan, and we compare these GSDs to the geometrical and physical properties of the landslides, such as their width, area, rock-type, drop height and estimated scar depth. All slides occurred in slightly metamorphosed sedimentary units, except two, which occurred in younger unmetamorphosed shales, with rock strength expected to be 3 to 10 times weaker than their metamorphosed counterparts. For 11 landslides we did not observe substantial spatial variations of the GSD over the

deposit. However, four landslides displayed a strong grain size segregation on their deposit with the overall GSD of downslope toe sectors three to ten times coarser than apex sectors. In three cases, we could also measure the GSD inside incised sectors of the landslides deposits, which presented percentiles three to ten times finer than the surface of the deposit. Both observations could be due to either kinetic sieving or deposit reworking after the landslide failure, but we cannot explain why only some deposits had a strong segregation. Averaging this spatial variability, we found the median grain size of the deposits to be

strongly negatively correlated to drop height, scar width and depth. However, previous work suggests that regolith particles and bedrock blocks should coarsen with increasing depth, opposite to our observation. Accounting for a model of regolith coarsening with depth, we found that the ratio of the estimated original bedrock block size to the $D_{50}$ of the deposit was proportional to the potential energy of the landslide normalized to its bedrock strength. Thus the studied landslides agree well with a published, simple fragmentation model, even if that model was calibrated on rock avalanches with larger volume and

stronger bedrock than those featured in our dataset. This scaling may thus serve for future modeling of grain size transfer from hillslopes to rivers, aiming to better understand landslide sediment evacuation and coupling to river erosional dynamics.

# 1 Introduction

Grain size is an essential parameter for understanding sediment transport and associated processes in river evolution or hazards related to sediment pulses. For geomorphologists, it is increasingly considered an important parameter for the long-term incision of bedrock streams (Sklar and Dietrich, 2001; Cook et al., 2013, 2014; Turowski, 2018), while it is an essential part of the sedimentological signal which is ultimately archived in stratigraphy (e.g., Armitage et al., 2011).

Still, there are many processes that control the grain size distribution (GSD) delivered to rivers, and they are poorly understood (Allen et al., 2015). In recent studies, models have been proposed describing how weathering in the critical zone reduces the original size distribution of bedrock before the grains reach the surface (Marshall and Sklar, 2012; Riebe et al., 2015; Sklar et al., 2017). However, in active orogens with high erosion rates (>0.5 mm/yr), landslides are likely the main providers of sediments to rivers (Hovius et al., 1997; Struck et al., 2015; Marc et al., 2019), and a large fraction of sediment may reach the river only partially weathered. Indeed, the limits of models predicting soil GSD and the need to account for GSD derived from fractured bedrock was recently shown (Neely and DiBiase, 2020), though the role of mass wasting in delivering and further fragmenting bedrock particles was not explored. In those settings, understanding and modeling the controls on landslide GSD should be an urgent goal, which has been addressed by few studies. Indeed, in contrast to river sediments for which many studies exist (e.g., Ibbeken, 1983; Whittaker et al., 2011; Chung and Chang, 2013; Guerit et al., 2014, 2018) landslide GSDs have rarely been measured, in part because the latter is considerably more difficult, time consuming and potentially dangerous than the former.

A few studies have measured and discussed in detail the GSD of some large historical landslide or rock avalanches, often putting forward to explain their data the various mechanisms of rock fragmentation and grain segregation (see Crosta et al., 2007, and references therein). Although interesting for their discussion in terms of rock mechanics, such case studies do not allow us to understand the regional variability of landslide GSDs, nor to derive physical scalings that could pave the way to model the GSD of material delivered to river networks by landslides. To our knowledge only seven studies reported detailed GSD measurements from multiple landslide deposits. A pioneering study reported the GSD from 42 landslide dams across the Appenines, with a discussion on the methods to derive the GSD but none on the controls of the GSD variability (Casagli et al., 2003). Locat et al. (2006) presented GSDs from nine large ($> 100\,\mathrm{Mm}^3$) rock avalanches from Canada and the Alps, including various rock types, and analyzed these in terms of potential energy and fragmentation theories (Locat et al., 2006). They found that the ratio of bedrock initial median block size, $D_i$ (estimated from fracture spacing), to the deposit median grain size, $D_{50}$, was proportional to the change in potential energy per unit of volume, $\rho g H$ (with $H$ the drop height of the center of mass, $\rho$ the rock density and $g$ the gravitational acceleration), normalized by the point-load strength of the bedrock, $\sigma_c$, measured with a point-load test performed on rock sample from the sites. Specifically their nine rock avalanches were best fit by a relation that could be recast as:

$$D_{50} = \frac{D_i}{k_1 \frac{\rho g H}{\sigma_c} - k_2} \tag{1}$$

Where $k_2 = 0.5$ is an empirical threshold for fragmentation, and $k_1 = 83.3$ is an empirical coefficient related to the conversion of potential energy into fragmentation energy and the effective breaking of particles. Thus, if the scaling is general for landslides, the deposit $D_{50}$ should increase with $D_i$ and $\sigma_c$ but decrease with $H$.

Subsequent studies often focused on the potential importance of landslide GSD for understanding sediment transport dynamics and the expected GSD at the outlet of basins, and reported GSDs in Nepal, Japan, California and Southern Italy (Attal and Lavé, 2006; Nishiguchi et al., 2012; Attal et al., 2015; Roda-Boluda et al., 2018). Several of them underlined, qualitatively, the factors influencing the GSDs such as the different lithological units (Attal and Lavé, 2006; Roda-Boluda et al., 2018), or the local hillslope gradient, as a control on the time spent in the weathering engine (Attal et al., 2015). Recently a study presented the GSD of seven medium size rockfalls in Spain, showing that the bedrock block size and the deposit GSD could be related through a fractal fragmentation model (Ruiz-Carulla and Corominas, 2020). They found that potential energy was a main control on the fragmentation, but no clear correlation with rock strength measures emerged. They did not compare their model and results to the simple scaling proposed by Locat et al. (2006). Thus, none of these more recent works has attempted to frame the landslide GSD in terms of the competition between fragmentation energy and source rock strength, and the scaling for large rock avalanches has not been reproduced on smaller, more frequent landslides.

Based on these studies, we formulate two hypotheses. First, we suggest that Eq (1) could be generalized to landslides of intermediate size and depth, and thus that landslide deposit $D_{50}$ should increase with rock strength, $\sigma_c$, and source material's median size $D_i$, but decrease with drop height, $H$. Second, we hypothesize that materials mobilized by shallow landslides coarsen with the landslide scar thickness, $T$, (i.e., $D_i$ increases with $T$), due to a reduction with depth of the fracture density of the bedrock (Clarke and Burbank, 2011) and/or of the degree of physical and chemical weathering experienced by particles (Cohen et al., 2010; Anderson et al., 2013; Sklar et al., 2017). Testing these hypotheses seems essential to pave the way towards geomorphic models accounting for the GSD of sediments transferred from hillslopes to rivers and from rivers to sedimentary basins (Allen et al., 2015; Sklar et al., 2017).

With these goals, and given the sparse amount of data on landslide GSDs, we performed detailed measurements on 17 recent landslide deposits in Taiwan. Taiwan is a prime example of an active mountain belt where landslides are the main supplier of sediment to rivers (Hovius et al., 2000) and where reports of river GSDs exist in the literature (Chung and Chang, 2013; Lin et al., 2014). Still, to our knowledge, comprehensive landslide GSD measurements are still lacking in Taiwan. Below we report our measurements and discuss the source of variability of the GSD within given landslides and across the whole dataset. Then, we discuss the validity of the two hypothesis stated above based on the GSD of these landslides. We end by discussing the implications in terms of caveats and opportunities for GSD sampling and implications for fluvial sediment transport.

## 2   Data and Methods

In this study we report original GSDs for 17 landslide deposits from Taiwan (Fig. 1), as well as basic landslide information that we use to discuss controls on the GSD (Table 1). We detail below how we constrained landslide characteristics and measured

GSDs for each deposit. Note that there are two landslides at the same site named LS-9o (for "old") and LS-9n (for "new") as the latter one appear to have happened after the former one (see Fig. 1D).

## 2.1 Landslide characteristics

To quantify the variability in landslide GSD and its controls, we have targeted landslides with a known triggering date, and covering a broad range of areas (40 m$^2$ to 0.1 km$^2$) and lengths (10 to 400 m). Except for four small landslides ($< 1000$ m$^2$), which were opportunistically sampled close to larger neighbouring ones, all landslides were targeted based on satellite imagery and chosen for the accessibility of their deposit.

Landslide type was difficult to assess, but most landslides could be called debris avalanches (Varnes, 1978), involving variable amount of regolith and bedrock, though LS-13 and LS-14 could also be called rock falls. LS-12, the largest event, may rather be a deeper rock slump, with moderate displacement, partly translational, partly rotational. Most landslides correspond to landslide polygons present in the Typhoon Morakot landslide inventory (Marc et al., 2018), and thus occurred in August 2009, about five and a half years before they were surveyed in March 2015. Other more recent landslides were dated based on the time-series of images available in Google Earth (see Table 1).

To assess variability in GSD independent of rock type, 13 out of 17 landslides were chosen in the same geographic area, on both sides of the southern section of Taiwan Central Range, in relatively homogeneous lithological units composed of slate and slightly metamorphosed sandstone (Fig. 1, Table 1). LS-1 and LS-15 also occurred on moderately metamorphosed units, on both sides of the northern part of the Central Range, in black schist and in metasandstone intercalated with slate, respectively. The two remaining landslides both occurred in unmetamorphosed units, made of alternating sandstone and shale for LS-10 which occurred in the emergent topography of Taiwan's southern tip, and in shales of the northwestern foothills for LS-16. In LS-16 many coarse rock fragments ($> 10 \, cm$) were crumbling when touched, highlighting the weakness of this rock compared to the other units.

In an effort to constrain the mechanical strength of these units, we refer to measurements reported for 128 samples from the Chenyoulan catchment, both for the Nanchuang/Nankang formation, which extends to the southern tip of Taiwan and contains LS-10, and for the metasedimentary units of the Shipachungshi formation, where LS-15 occurred (Lin et al., 2008). The unconfined compressive strength of Nanchuang sandstone ranged from 29 to 117 MPa (mean of 70 MPa), while the Shipachungshi metasandstone ranged from 45 to 179 MPa (mean of 100 MPa) (Lin et al., 2008). However, in the Nanchuang formation, sandstones are alternating with weak shale (strength below 10 MPa), with an equal proportion of each. In contrast, the Shipachungshi metasandstone is intercalated with less frequent slates, often stronger than the Nanchuang shales (Lin et al., 2008). These measurements clearly make the case for highly variable rock strength and are far from encompassing the potential diversity of rock-type sampled by the studied landslides. Our goal here is not primarily to constrain the rock strength of individual landslides, but to estimate the relative strength of diverse units. Based on the measurements reported above we make two assumptions. First, that the shales and sediments hosting LS-16 and LS-10 may be 7 to 13 times and 2 to 4 times weaker, respectively, than the metasediments hosting LS-15. Second, that the slates and metasandstones in the Lushan

formation have similar strength to the ones in which LS-15, as well as LS-1, occurred and thus that these landslides can be compared without normalizing for strength.

Geometric landslide metrics were obtained from high resolution satellite imagery available in Google Earth except for four deposits (LS-4, 9n, 13 and 14), which were too small to be clearly distinguished on the imagery, and which had their dimensions approximated from field observations only, using a laser range-finder. Area was obtained by hand mapping the whole disturbed zone on the imagery. Length refers to the downslope length between the highest and lowest point of the polygon. The elevation difference between these two points, estimated from the elevation data of Google Earth (in Taiwan mostly 30 m SRTM, predating all of the studied landslides), defined the maximum drop height. Physically, the potential energy change in Eq (1) is related to the displacement of the center of mass. Thus, we also estimate a length and drop height from the center of mass of the scar to that of the deposit, estimated from Google Earth imagery. An estimate of pre-failure scar gradient could be derived from the scar's approximate length and height difference.

Beyond plan-view metrics, we must also estimate landslide scar volume in order to constrain the landslide scar depth. For a few landslides the deposit volume could be approximated as a fraction (a quarter to a half) of a cone, for which a volume estimate could be obtained as $\pi R^2 h/3$ with $R$ and $h$ the approximate radius and height of the cone which were estimated in the field. This simplified geometry was only suitable for LS-3, 4, 7, 9n, 10, 11, 13 and 15, and yielded only a first order "field volume" estimate (Table 1). For the other deposits we had to rely on scaling relationships between scar area and volume, with the additional complexity that the lower extent of the scar area could not be clearly assessed in most cases, even with high resolution imagery. Thus, assuming the mean scar aspect ratio from a global database apply to the surveyed landslides (Domej et al., 2017), we estimated scar areas as $A_s = 1.5W_s^2$, with $W_s$ the scar width obtained by measuring the extent of the landslide in the direction orthogonal to flow, in the upper part of the failure only. With $A_s$, we estimated a maximum and minimum landslide volume using empirical scaling relationships of the form $V = \alpha A_s^\gamma$ , with different parameters values assuming the scar was mobilizing soil or bedrock, respectively. We used $\gamma = 1.262 \pm 0.009$ and $log10(\alpha) = 0.649 \pm 0.021$ for landslides in soil, and $\gamma = 1.41 \pm 0.02$ and $log10(\alpha) = 0.63 \pm 0.06$ for landslides in bedrock (Larsen et al., 2010). Then we derived the upper and lower estimates of landslide mean scar depth as the ratio of volume to scar area for bedrock and soil, respectively. Although approximate, this scaling is still preferable to using total landslide area, as it removes bias in volume estimates associated with variable runout length, difficult to constrain from field or satellite observations (e.g., Marc et al., 2019). Where available, the volumes estimated from the field mostly fall within the bracket of the volumes estimated from global scaling relationships (Fig. S1), lending some support to this approach.

Still, for the deposits where we could not obtain a field estimate, better constrained volume estimate could be obtained by choosing one of the two scaling relationships. We note that the field volume estimate of LS-3 and LS-4 is similar to the estimate from the soil scaling relationship (Fig. S1). This is consistent with the observation that they were composed of rock debris with a yellowish color that indicated advanced weathering, and contained fresh vegetation debris (see Fig. 1). For LS-7 and LS-11, which were clearly involving mostly fresh bedrock, the field volume estimates match better with the bedrock scaling relationship (Fig. S1). Thus, where field volumes were lacking we used the bedrock estimate for the largest landslides ($W_s > 50$) within which the rock looked mostly fresh (i.e., LS-1, 2, 8, 12). Some other landslides (LS-5, 6, 9o, 16), featured a

mixture of soil and rock material. Consequently, we used the average of the soil and bedrock scaling to estimate their volume. The best estimate for each landslide was divided by its scar area to obtain an estimate of scar thickness (Table 1).

## 2.2 Grain size counting

GSDs were obtained using grid-by-number sampling, following established protocols developed for measuring riverine GSDs (see Kellerhals and Bray, 1971) and subsequently applied to landslide deposits (Casagli et al., 2003; Attal and Lavé, 2006). We extended survey tapes along an elevation contour over a substantial portion of the deposit width (10 to 50 m) and sampled grains along the tape at a constant interval, recording the size bin of the b-axis, measured with rulers. We used bins following a half Phi scale (power of 2 by 0.5 increments) with the smallest bin encompassing all grains finer than 2 mm. When grains
could not be moved, we considered the smallest of the two visible axes as the b-axis. The sampling interval was 0.5 m in most cases but was adjusted to 1 meter for deposits where many meter-scale boulders where present (LS-2s, LS-13, LS-14) to avoid having to count many grains several times. Then we moved the line in parallel, upslope by one to a few meters depending on the deposit's dimensions and local topography, and repeated the counting. Most slides were sampled with 6-10 survey lines allowing us to cover a substantial fraction of the deposit (often 30 to 60%), with total counts typically including $200 - 400$
individual grains. This approach also allowed us to sample different sections of the deposit when a spatial segregation was visible (LS-3, LS-8, LS-9n, LS-10), and to quantitatively assess this spatial variability in grain size (see Ruiz-Carulla et al., 2015).

Specifically, we observed and measured spatial variations of the GSD on the surface deposit for four landslides (LS-3, LS-8, LS-9n, LS-10). In addition to the surface GSD, for three landslides (LS-2, LS-5, LS-8) we also measured the GSD of a section
of the deposit likely to represent the interior of the deposit. Below we explain how we could measure independently various GSD on these landslide deposits. For LS-8 and LS-2, we counted grains on the vertical banks of a 2 m deep erosional gully incising the deposit, and on a debris fan next to and below the road that had been cleared from the deposit, respectively. Thus the former case allowed us to survey the internal GSD in place, while the latter likely represents a remixing from surface and internal parts, and thus must be closer to the inner GSD than what would be derived from surface measurements only. Note
that for LS-2 the only undisturbed deposit was the one in the transport channel, where a carapace (a layer of very coarse grains, Crosta et al. (2007)) seems to have formed (Fig. S2). Finally, on LS-5 we measured separately a debris fan and the terminal section of a channelized deposit which was visibly coarser (Fig. S3). In this case, given the age of the deposit and its direct contact with the floodplain, it is plausible that the deposit was partly eroded and the fan may be a mixture of internal and superficial material, whereas the higher up channel section may be more representative of the original surface of the deposit.
This will be further discussed when discussing segregation, below. In any case, to differentiate between GSD considered to represent the interior or surface of the deposit we add the letter "i" or "s" after the name, respectively (Fig. 2).

Additionally, on the deposit of LS-7 we could distinguish by visual inspection grains made of slate, which were dark, elongated and without visible internal structure, from grains made of metasandstone which were lighter, more cubic and with visible internal grains. We have counted them separately as we found them over the deposit. In many other deposits a large majority of

**Table 1.** Landslide characteristics for the 17 surveyed deposits. Asterisks indicate landslides for which the geometry was estimated in the field rather than from satellite imagery. Due to its complex displacement LS-12 has large uncertainties on the displacement of its center of mass. Bsc=Black schist (Tananao Fm); Sl/Sd= Slate/Sandstone (Lushan Fm); Sh/Sd=Shale Sandstone (NanChuang Fm) ; Msd=Metasandstone (ShihPachungshi Fm); Sh=Shale (Chinshui/Cholan Fm). For the subsectors: U=Up; M=Mid; T=Toe; I=In; C=Channel; NA: Not Applicable.

| Landslide Number | 1 | 2 | 3 | 4* | 5 | 6 | 7 | 8 | 9o | 9n* | 10 | 11 | 12 | 13* | 14* | 15 | 16 |
|---|---|---|---|---|---|---|---|---|---|---|---|---|---|---|---|---|---|
| Total grain count | 303 | 640 | 598 | 393 | 289 | 196 | 362 | 978 | 178 | 334 | 268 | 344 | 494 | 353 | 141 | 402 | 166 |
| Subsectors counts | NA | I(496)/C(144) | U(206)/M(204)/T(188) | NA | C(124)/T(165) | NA | Sl(167)/Sd(195) | U(334)/T(414)/I(230) | NA | U(108)/M(121)/T(105) | U(165)/T(203) | NA | NA | NA | NA | NA | NA |
| Landslide Length, Lmax, (m) | 192 | 400 | 100 | 25 | 125 | 25 | 135 | 330 | 120 | 15 | 60 | 270 | 330 | 20 | 10 | 310 | 160 |
| Horizontal displacement of the center of mass, L, (m) | 100 | 330 | 70 | 15 | 95 | 15 | 90 | 250 | 85 | 8 | 40 | 220 | 120 ±50 | 10 | 5 | 240 | 90 |
| Scar Width, $W_s$, (m) | 80 | 90 | 60 | 20 | 30 | 15 | 35 | 70 | 25 | 5 | 20 | 60 | 260 | 15 | 10 | 180 | 80 |
| Area, A, ($m^2$) | 13171 | 33583 | 7801 | 756 | 2228 | 538 | 6115 | 14523 | 1834 | 45 | 921 | 18196 | 114454 | 300 | 150 | 58751 | 15214 |
| Longitude (°) | 121.415 | 120.948 | 120.896 | 120.894 | 120.857 | 120.857 | 120.899 | 120.852 | 120.857 | 120.857 | 120.775 | 120.657 | 120.679 | 120.681 | 120.681 | 120.902 | 120.833 |
| Latitude (°) | 23.734 | 22.597 | 22.516 | 22.514 | 22.439 | 22.435 | 22.508 | 22.438 | 22.435 | 22.435 | 22.135 | 22.444 | 22.519 | 22.517 | 22.516 | 23.609 | 24.286 |
| Center of Scar Elev. (m) | 560 | 380 | 200 | 175 | 300 | 285 | 480 | 420 | 305 | 282 | 240 | 265 | 460 | 415 | 430 | 1050 | 450 |
| Center of Deposit Elev. (m) | 485 | 130 | 160 | 160 | 240 | 270 | 410 | 300 | 275 | 275 | 225 | 155 | 390 | 405 | 410 | 870 | 405 |
| Drop of the center of mass, H, (m) | 75 | 250 | 40 | 15 | 60 | 15 | 70 | 120 | 30 | 7 | 15 | 110 | 70 ±30 | 10 | 20 | 180 | 45 |
| Maximum Drop, Hmax, (m) | 130 | 320 | 60 | 20 | 70 | 25 | 120 | 170 | 40 | 12 | 25 | 180 | 180 | 20 | 25 | 260 | 70 |
| Scar Gradient | 1 | 0.9 | 0.6 | 0.6 | 0.9 | 1 | 1.2 | 0.8 | 0.8 | 0.8 | 0.6 | 1 | 0.55 | 2.15 | 2.15 | 1 | 0.6 |
| Volume (Bedrock scaling), V, ($m^3$) | 96606 | 134664 | 42922 | 1937 | 6078 | 861 | 9388 | 66293 | 3635 | 39 | 1937 | 42922 | 3305783 | 861 | 274 | 950949 | 96606 |
| Volume (Soil scaling), V, ($m^3$) | 15405 | 20728 | 7461 | 468 | 1301 | 227 | 2904 | 11003 | 822 | 14 | 468 | 7461 | 362004 | 118 | 82 | 118894 | 15405 |
| Field Volume, ($m^3$) |  |  | 16000 | 850 |  |  | 12500 |  |  | 24 | 750 | 43000 |  | 600 | 115 | 550000 |  |
| Best Volume (see Methods), ($m^3$) | 96606 | 134664 | 16000 | 850 | 3690 | 544 | 12500 | 66293 | 2228 | 24 | 750 | 43000 | 3305783 | 600 | 115 | 550000 | 56000 |
| Thickness (Bedrock scaling), (m) | 10.06 | 11.08 | 7.95 | 3.23 | 4.5 | 2.55 | 5.1 | 9.02 | 3.88 | 1.04 | 3.23 | 7.95 | 28.11 | 2.55 | 1.83 | 19.57 | 10.06 |
| Thickness (Soil scaling), (m) | 2.43 | 2.58 | 2.09 | 1.18 | 1.46 | 1.02 | 1.6 | 2.27 | 1.33 | 0.57 | 1.18 | 2.09 | 4.66 | 1.02 | 0.82 | 3.7 | 2.43 |
| Best Thickness (see Methods), T, (m) | 10.06 | 11.08 | 2.96 | 1.42 | 2.73 | 1.61 | 6.8 | 9.02 | 2.37 | 0.64 | 1.25 | 7.96 | 28.11 | 1.78 | 0.77 | 11.32 | 5.83 |
| Occurrence Year (Jul-Sep) | 2013 | 2012 | 2014 | NA | 2009 | 2009 | 2009 | 2009 | 2009 | NA | 2013 | 2009 | 2009 | NA | NA | 2014 | 2013 |
| Rock type | Bsc | Sl/Sd | Sl/Sd | Sl/Sd | Sl/Sd | Sl/Sd | Sl/Sd | Sl/Sd | Sl/Sd | Sl/Sd | Sh/Sd | Sl/Sd | Sl/Sd | Sl/Sd | Sl/Sd | Msd | Sh |

grains either looked fairly homogeneous, or because of lack of time their systematic counting within different rock types could not be done. To study the variability between various landslides we obtained an overall surface GSD by summing the grain counts from both lithologies of LS-7 and from the different sectors of the deposits with spatial segregation. For these cases we did not use an area weighted sum (Ruiz-Carulla et al., 2015) because the upper, middle (when differentiated) and lower sections of the deposits represented roughly similar proportion of the surface of the deposits, and we obtained count variations below 10% from the different subsections (Table 1, Fig 3). When we measured both internal and superficial GSDs we had to select one of them as representative for the comparison to other landslide properties (see section 3.2).

## 3 Results

### 3.1 Landslide grain size distributions and their internal variability

Before averaging spatial variability, the landslide GSDs have 50th and 84th percentiles ranging from about 15 to 200 mm and about 60 to 600 mm, respectively. This is consistent with the range of observations from previous studies, except the large rock avalanches from Locat et al. (2006) and the volcanic rock avalanches from Crosta et al. (2007), which were about 10 times coarser and finer, respectively, than all other studies. LS-2s and LS-16 are much coarser and finer than the rest of the studied landslides, respectively. Interquartile ratios vary between 3 and 15, but we note that 13 out of 20 GSD have an interquartile ratio of 3 to 6, while only LS-1, 3, 5i, 5s, 8i, 15 and LS-16 have larger spreads (Fig. 2). All distributions seem unimodal, except LS-16 with more than 40% of the grains finer than 2 mm, likely containing a second, sub-millimetric mode that could not

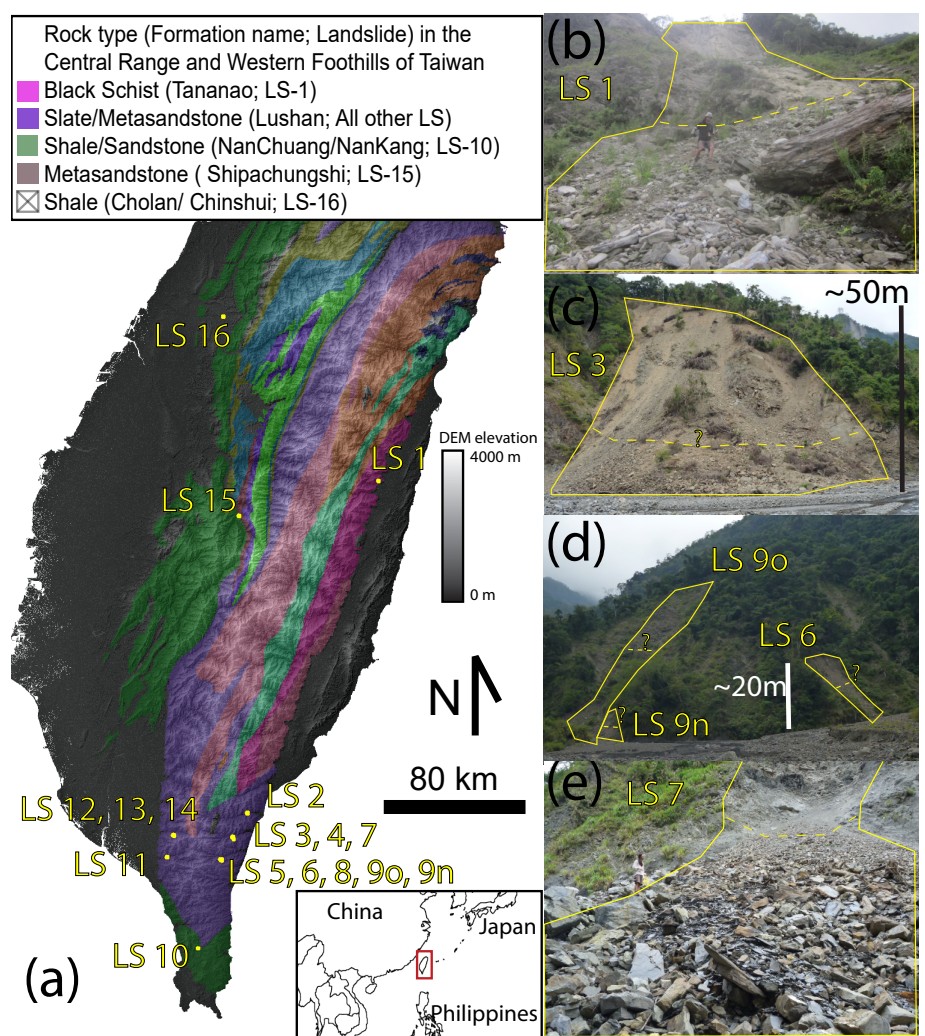

**Figure 1.** (a) Hillshaded elevation map of Taiwan, with the main lithological units of the central range (based on the geological map of Taiwan available from Taiwan Central Geological Survey) and the locations of the 17 sampled landslides deposits. (b-e) Pictures of some sampled landslides, where the yellow line is the approximate contour of the landslide (sometimes going beyond the pictures), and the dashed line indicates the transition from deposit to scar (it is only tentative when associated with "??"). In (b) and (e) the lead author is standing on the deposit for scale.

be constrained by our methods. Grain size distributions can often be well described by a Weibull or Lognormal distribution (Ibbeken, 1983). For the studied landslides, eight GSDs are better fit (according to both Kolmogorov-Smirnov and Anderson-Darling statistics, Stephens (1974)) by a Weibull distribution (LS 2s, 3, 4, 5s, 5i, 6, 9n, 9o, Fig. 2B, S4), while all others are better fit by a log-normal distribution (Fig. 2A, S5). Note that LS-16 is poorly fit by both distributions. These two subgroups
5    imply that we cannot prescribe a given distribution form to model landslide GSDs, and future work may aim at understanding

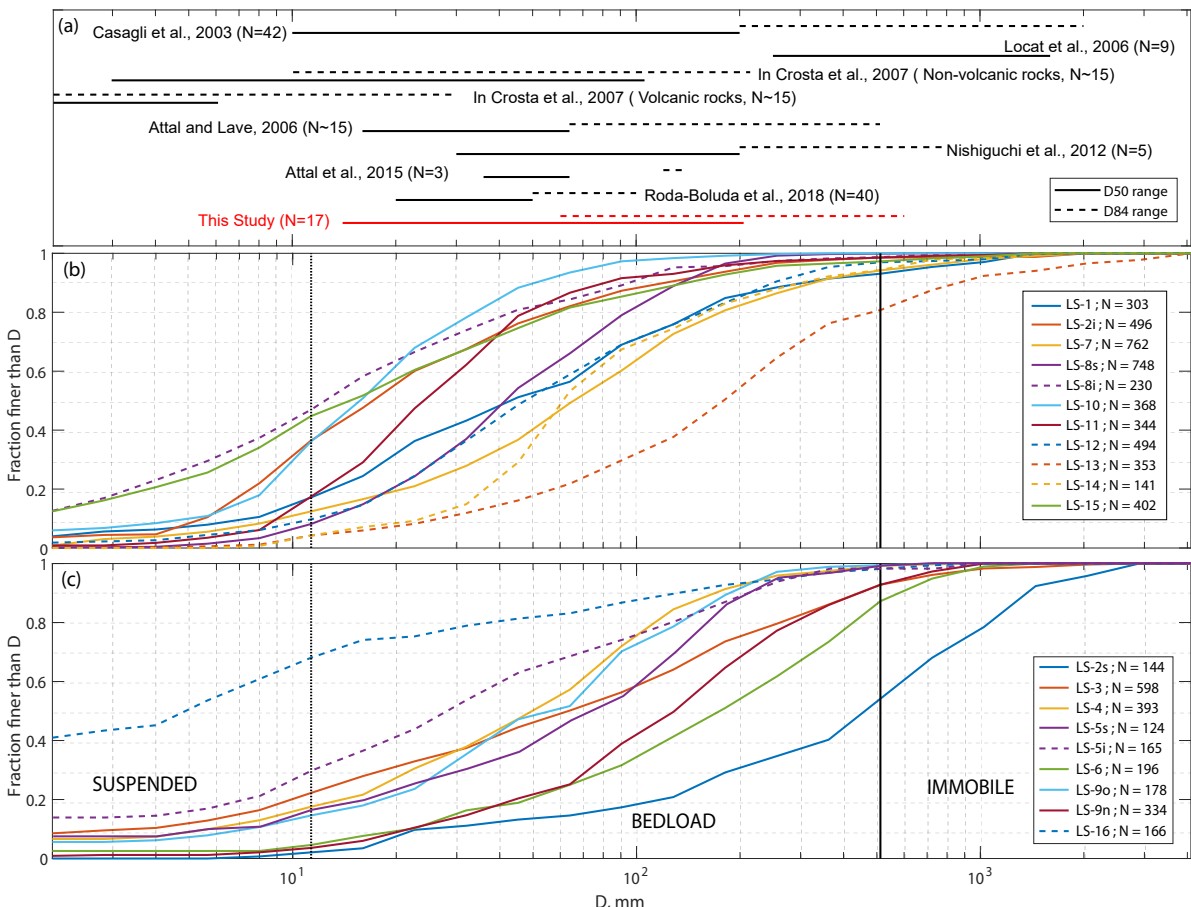

**Figure 2.** (a) Range of $D_{50}$ and $D_{84}$ for various studies, excluding the coarsest and finest distributions of each study. (b) and (c) Cumulative distribution function for the 17 sampled landslide deposits. Note that we show surface and inner distributions for LS-5, LS-8 and LS-2, and thus 20 distributions in total. For visibility some lines are dashed and the distributions are shown in two panels. Vertical lines are approximate boundary for grain transport by suspension and bedload, for a flood associated with fluid shear stress of 220 Pa (see section on Implications for sediment transport).

why some landslide GSDs obey one or the other distribution. In any case, we refrain from using distribution parameters and will continue to discuss results based on empirical descriptors (i.e., median, interquartile ratio).

GSDs within a single landslide deposit were often heterogeneous, in one case associated with differences between grains of
5   different rock types (slate and sandstone in LS-7), while in seven other cases associated to spatial variability across the land-slide(Fig. 3, S6). For LS-7, the slate pieces have grain sizes about three times smaller than the sandstone for a given quantile of the GSD, with a similar distribution shape. The slate grains were typically elongated platelets (i.e., with a-axis about three times longer than b-axis, and c-axis much smaller than b-axis), while the sandstone grains were cubic and slightly more abundant

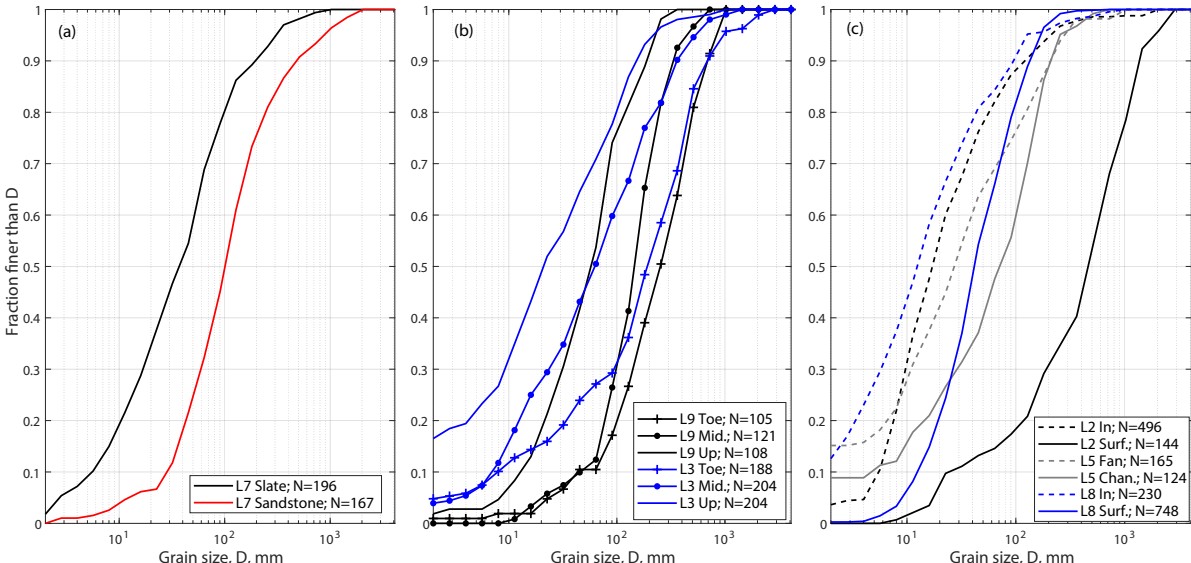

**Figure 3.** Examples of three types of heterogeneity in grain size distributions within the same landslide deposit. (a) Lithological difference within the deposit of LS-7 with sandstone grains coarser than slate grains. (b) Downslope differences between the upper, middle and lower part of the deposits of LS-3 and LS-9. (c) Difference between the surface and inner part of the deposits of LS-2, LS-5 and LS-8.

than the slate grains (N=196 vs N=167). We observed with the naked-eye downslope segregation, i.e., an increase in sediment coarseness from the apex to the toe of the deposits (Ruiz-Carulla et al., 2015), in four cases. The strongest segregation occurred in deposits LS-3 and LS-9n, where the upper part of the deposits have grains 5-10 times finer than the lower part of the deposit, without changing substantially the shape of the distribution. Deposits LS-8 and LS-10 exhibited a more subtle segregation with

5    the upper part of the deposits having distributions finer by a factor 1.5-2 in comparison to the toe of the deposits (Fig. S6). The toe of LS-10 also displays $D_{50}$ and $D_{84}$ twice as coarse as at its apex, consistent with other cases, but also has more fine grains, with about 10% of grains finer than 2 mm against less than 5% at the apex.

     In two cases, we could separately measure the superficial and internal GSD. For LS-8 we observed that the superficial GSD

10    had $D_{16} = 20$ mm, $D_{50} = 40$ mm and $D_{84} = 120$ mm while the internal GSD had $D_{16} = 3$ mm, $D_{50} = 10$ mm and $D_{84} = 50$ mm. There the superficial deposits had almost no fine sediment below 2 mm, whereas the internal body had more than 10% of fine sediments. Thus, the internal GSD had quantiles 10 to 20 times finer than the channel carapace, the largest difference observed in terms of internal variability. Note that the carapace had also a coarser GSD than any other measured landslide deposit in our study. In spite of this large difference, we note that the internal GSD still had only about 3% grains finer than 2

15    mm. These two examples clearly show that the superficial GSD can be substantially different from the internal GSD, both in terms of fine grains ($< 2$ mm) but also for coarse to very coarse grains (10 to 100 mm).

Last, in the case of LS-5, it is not entirely clear if the two distributions represent vertical segregation or superficial spatial

variability. Given that the fan has a $D_{16} = 4$ mm and a $D_{50} = 20$ mm, about three times finer than in the channel, but an almost identical $D_{84}$ of around 200 mm, we consider it to likely be an internal or mixed GSD.

## 3.2 Relationship towith landslide properties

The percentiles of the GSDs are highly correlated with linear correlation coefficients having $R^2 > 0.9$ between $D_{50}$ and $D_{16}$, $D_{25}$, $D_{75}$, $D_{84}$ and $D_{90}$ (Fig. S7). We note that more scatter is present for $D_{16}$ and $D_{90}$, suggesting that $D_{50}$ is a good proxy for the bulk of the distributions, but does not completely capture the variability in their tails. The interquartile ratio (here $D_{75}/D_{25}$) which characterizes the span of grain size in the distribution, is also independent of the other percentiles (Fig. S8). To compare with landslide properties we used a single GSD for each landslide. For lithological variations or spatial variability on the surface we averaged the various GSDs from each landslide For the three landslides for which we have both an internal and superficial grain counting, we identified a single GSD that is most relevant for comparison with the other deposits. For LS-8 we considered the GSD at the surface, to be consistent with all other cases. For LS-5 we considered the coarser distribution from the channel as more representative of the surface deposit. If the fan of LS-5 is representative of the deposit and if the channel is coarser because of some spatial segregation, the percentiles of its GSD would be about two to three times finer than the ones we have selected (Fig. 3). In contrast for LS-2 we considered the internal GSD, because the superficial measurement recorded only what seems to be a carapace that over-represents coarse grains. Indeed, with about half of the distribution made of boulders ($> 0.5$ m) we consider that segregation to be exceptional within our dataset. Thus, in using LS-2i to study inter-event variability we assume that none of the other landslides had a carapace with similar strong sorting. Nevertheless, LS-2i percentiles may be biased towards finer grains, when compared to surface deposits of the other landslides, and we assume the bias could be up to a factor of two to three, based on LS-5 and LS-8 (Fig. 3).

According to Eq (1), and assuming similar initial regolith material for all landslides, $D_{50}$ should decrease linearly with the ratio of drop-height, $H$, to bedrock strength, $\sigma_c$. Indeed, log-transforming and fitting $D_{50}$ against $H$ for the 15 deposits from the metasedimentary units, which we expect to have relatively similar strength, we obtain $R^2 = 0.71$ and a power-law exponent of -0.64 (Fig. 4). Including LS-10 and LS-16 from weaker units yields a substantially poorer fit ($R^2 = 0.31$). Thus, to account for their weaker strength, we rescaled these two landslide drop heights by a factor of 3 and 10, respectively, reflecting the central values estimated from strength measurements (see methods). Then, we obtained a correlation coefficient for all of the landslides ($R^2 = 0.71; N = 17$) and a best-fit power-law exponent of -0.78. For landslides in metasedimentary units, $D_{50}$ is also negatively correlated (with a larger scatter $R^2 = 0.5 - 0.55$) to landslide size metrics (area, width, volume, depth). However, we note that, for this dataset, these metrics are also strongly correlated with the drop height ($R^2 = 0.56 - 0.66$, Fig. 5). Given that we would expect larger and deeper landslides to mobilize fresher and coarser grains, and thus to have positive corelation with $D_{50}$, these negative correlations may simply reflect the fact that in our dataset deeper landslides have a larger drop height, and that the effect of drop height in decreasing $D_{50}$ is more important than the effect of landslide size in sourcing coarser material.

For the spread of the distribution, characterized by the interquartile ratio, we did not find any substantial correlation with any

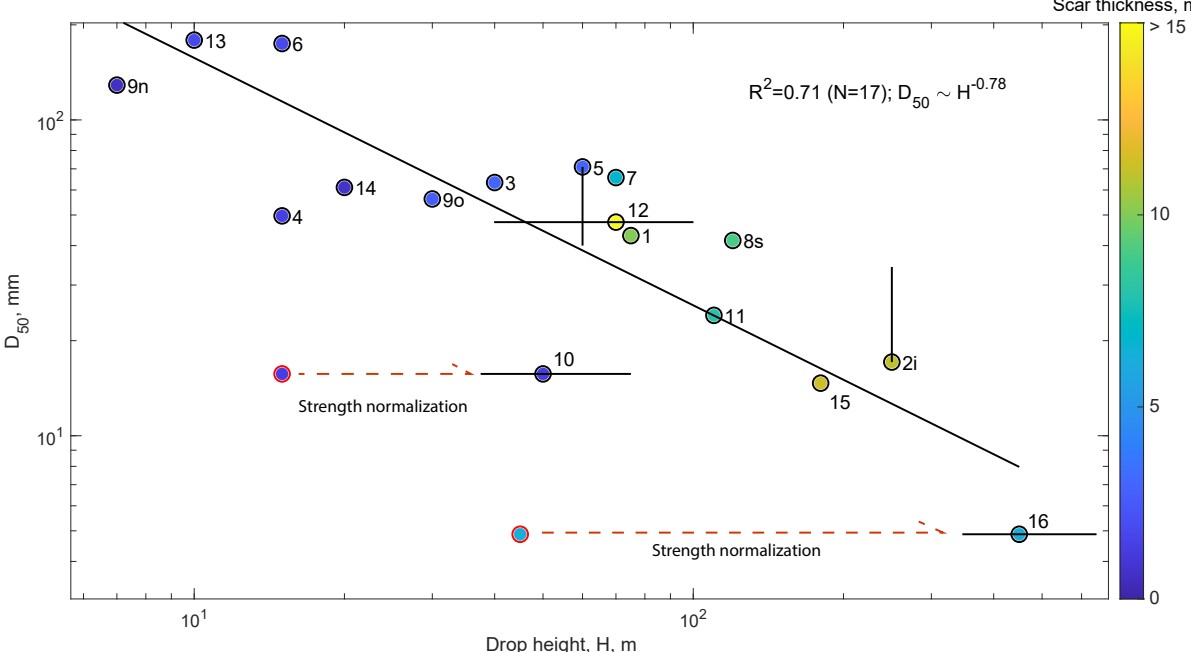

**Figure 4.** $D_{50}$ for the 17 landslides (colorcoded by scar thickness) of this study against the drop height of their center of mass. Results are similar for maximal drop height. The best-fit (solid black line) and $R^2$ only consider the drop height rescaled by the landslide rock mass strength (black circles). The red circles show the original drop height for these two landslides. Vertical bars show a factor of two uncertainty for LS-5s and LS-2i, for which there may be vertical segregation (see text). Horizontal error bars represent uncertainties on the drop height of LS-12 and the strength normalization of LS-10 and LS-16 (see methods).

of the landslide variables. Even the rock type (or rock strength) does not seem to have an impact on the GSD spread, with several landslides in metasedimentary rocks with very large spreads (LS-3, LS-5, LS-1), while the two landslides in non-metamorphosed units are on both ends of the spectrum. Thus, more data are needed to understand the spread of the landslide GSDs.

## 4    Discussion

In the following discussion, we propose that the variability of landslide $D_{50}$ can be reconciled with the fragmentation scaling of Locat et al. (2006) (i.e., $D_{50}$ decreases with the ratio of drop height to bedrock strength, as in Eq (1)), when accounting for regolith coarsening with depth (e.g., Cohen et al., 2010). Then we detail processes that may lead to grain size segregation within a given deposit and practical implications for sampling. Last we explore the implication of the measured GSD on sediment transport and evacuation from river channels.

Before turning to these points, we recall that some of our variables that are important for modelling landslide $D_{50}$ (section

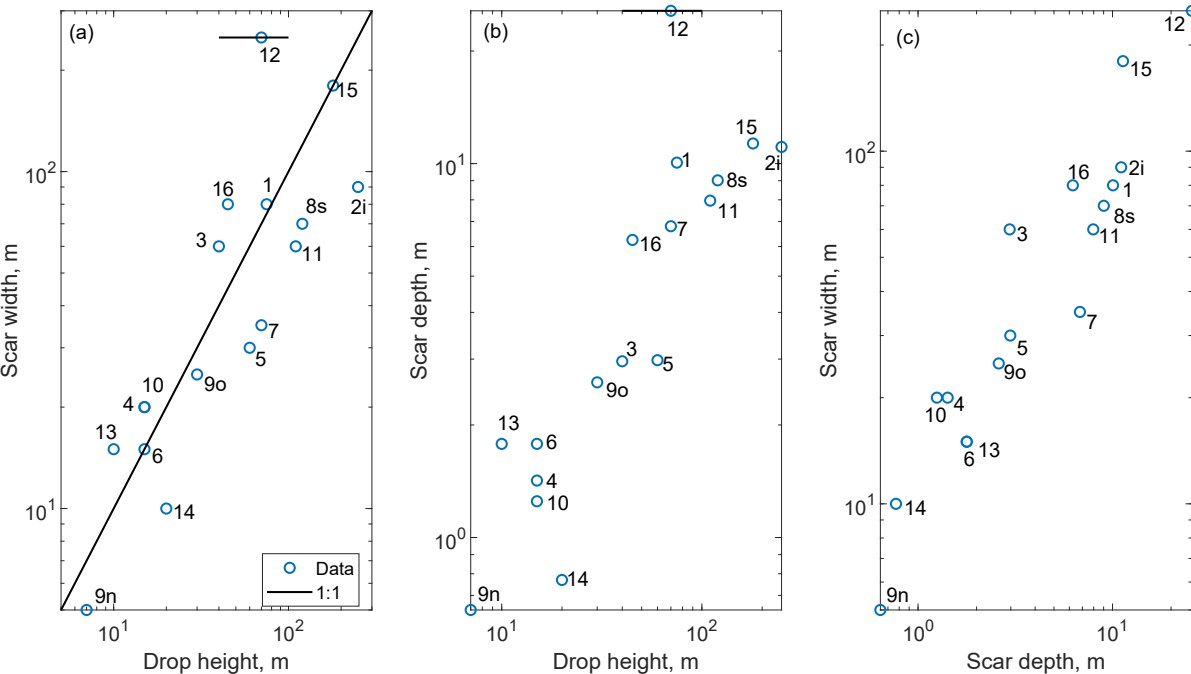

**Figure 5.** Correlation between the principal geometric dimensions of the surveyed landslides, drop height, scar width and scar depth. Note that scar depth is computed directly from scar width for all the landslides for which we do not have volume estimates from the field (i.e. all except LS 3, 4, 7, 9n, 10, 11, 13, 14 and 15, see Methods).

4.1) are only first order estimates, if available at all. First, we rely on rough estimates of the landslide geometry (drop height, volume, depth, etc). Better characterization in the field would have required more field work, and was limited by accessibility, or elaborate construction of DEM based on LIDAR or drone photogrammetry, which is difficult to perform and limited by the lack of accurate pre-failure DEMs. Thus for the sake of this first study we think that having consistent, first-order estimates of these metrics is sufficient to test the dependence of the $D_{50}$ on landslide geometry. The difficulty in accessing many scars, as well as the mixed origin (i.e., weathered regolith and bedrock) of several landslide sources also meant that in practice we could not measure the source materials median grain size, $D_i$. Nevertheless, we propose below that variability in $D_i$ may be captured with existing weathering models.

### 4.1 The importance of fragmentation and source material initial grain size

Here we discuss the hypothesis that Eq (1), proposed and validated by Locat et al. (2006) for large rock avalanches, can also be used for smaller, shallower landslides made of a mixture of regolith and bedrock. For the 17 Taiwanese landslide in our study we found that within a given lithology, drop height seems to be a first order control on the landslide deposit median grain size (Fig. 4). We also found that by rescaling the drop height by their weaker rock strength, LS-10 and LS-16 were consistent with the trend defined by the stronger metamorphosed units. These observations qualitatively agree with Eq (1), but quantitatively,

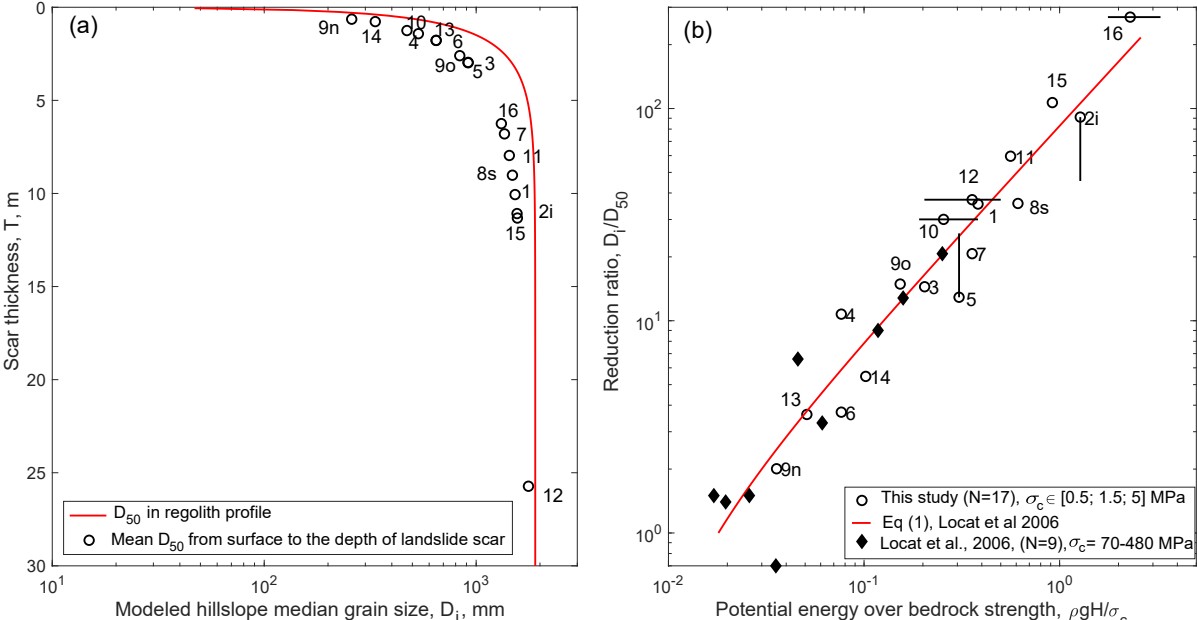

**Figure 6.** (a) Grain size as a function of depth in the regolith inspired by the weathering model by Cohen et al. (2010), used to estimate the original median grain size, $D_i$, mobilized by landslides with different thicknesses. (b) Reduction ratio (using the modeled $D_i$ from (a)) against the potential energy normalized by point load strength estimate for the 17 landslide deposits of this study. Error bar as in Fig. 4

the best fit between $H$ and $D_{50}$ was not linear, but a sub-linear power-law. Given that we observe that $H$ and the landslide scar thickness, $T$, are correlated in our surveyed landslide (Fig. 5) this discrepancy with Eq (1) could be resolved if $D_i$, which we could not measure, is increasing with $T$. Models describing the size of particles in a soil or regolith predict upwards fining of grains from the bedrock to the surface due to an increase in the degree of both physical and chemical weathering (Cohen

5 et al., 2010; Anderson et al., 2013; Sklar et al., 2017). In bedrock, fracture density estimated from seismic wave refraction was also found to decrease non-linearly from the surface to a depth of 5-10 m (Clarke and Burbank, 2011). Given soils are often thin in Taiwan, and represented a small proportion of the mobilized material, we consider physical weathering is likely dominant. Here we consider that the physical weathering rate (i.e., the rate of particle breakdown) can be modelled with an exponential decay from the surface, with a characteristic length scale of $\lambda = 2$ m, consistent with previous modeling (Cohen

10 et al., 2010; Anderson et al., 2013). Assuming the regolith median grain size at a depth $z$, $D_r(z)$, to be proportional to the integral of the weathering rate, we modeled it as $D_r = D_b(1 - exp(-z/\lambda))$ where $D_b$ is the unweathered bedrock block size, producing a rapid variation near the surface consistent with published models for physical weathering (Cohen et al., 2010; Anderson et al., 2013). Averaging $D_r$ from 0 to $T$, the mean scar thickness, and choosing a value for $D_b$ we can obtain $D_i = D_b(1 - \lambda/T(1 - e^{-T/\lambda}))$ for each landslide (Fig. 6a).

15 To compare with the prediction of Locat et al. (2006) (i.e.,Eq (1)), we have to assume a value for $D_b$ and use point load strength values ($\sigma_c$) of 0.5 MPa for LS-16, 1.5 MPa for LS-10 and 5 MPa for the other landslides in metasediments. These

point load measurements are consistent with the typical unconfined compressive strength of intact rock (Lin et al., 2008) after dividing them by 20 (Chau and Wong, 1996). With these strength values and assuming $D_b = 1900$ mm, the ratio of the modeled $D_i$ to the measured $D_{50}$ agrees with Eq (1) ($R^2 = 0.9$), even though its coefficients ($k_1$ and $k_2$) were calibrated on rock avalanches with rock strength and median grain size orders of magnitude larger than the ones from this study (Table 1, Fig. 2a, 6b). All of the surveyed landslides are within a factor of two from the predictions of Eq (1), even LS-12 which likely had a different deformation style than the other landslides. This good agreement is not very sensitive to how we estimate landslide volume and scar thickness (Fig. S9). Note that by assuming $D_b = 1900$ mm we have matched the $D_{50}$ of surface deposits, which may slightly over-estimate the representative grain size relative to the whole landslide deposit, because of kinetic sieving or fine removal by surface runoff (Fig. 3C and discussion below). Based on LS-5 and 8, the inner $D_{50}$ may be 2-3 times finer than its surface counterpart, and thus field measurement of regolith and bedrock GSD may need to be compared to a model with $D_b \sim 600 - 1000$ mm, to which uncertainty on rock strength (i.e., probably a factor of two) should also be added. Still, although uncertain, these latter values are intermediate between fracture measurement on surface outcrops in the US (Neely et al., 2019; Verdian et al., 2020) ranging from 10 to 400 mm, and the bedrock block size measured by Locat et al. (2006) on the scar of large rock avalanche, ranging from 600 to more than 10,000 mm. $D_b \sim 600 - 1000 \; mm$ also matches quite well the range of $D_{50}$ to $D_{75}$ (450 to 900 mm) found in the carapace of LS-2 which may be a fair first order estimate of the original bedrock block size (Crosta et al., 2007). We also note that LS-10 and 16, which occurred in weaker bedrock, may be expected to have a finer $D_b$ than the other slides. We are not able to constrain this but note that even with a $D_b$ three times finer, these two slides would be only a factor of 2 below the predicted $D_{50}$ (Fig. 6).

We conclude this section by underlining that more measurements, especially of source rock block size and strength, are needed to fully demonstrate the applicability of the fragmentation theory presented by Locat et al. (2006). Still, we suggest that such fragmentation theory is applicable to understand and predict landslide GSD in a wide range of contexts, at least for rock, soil and mixed avalanches and generally disrupted slides, which are the most commonly triggered (Keefer, 1984). Further, our observations suggest that Eq. (1) can be generalized to account for an exponential reduction of regolith grain size towards the surface (Cohen et al., 2010; Anderson et al., 2013), yielding:

$$D_{50} = \left( 1 - \frac{\lambda}{T}(1 - e^{-T/\lambda}) \right) \frac{D_b}{k_1 \frac{\rho g H}{\sigma_c} - k_2} \tag{2}$$

where $D_i$ has been replaced by a term depending on the "fresh" bedrock median size, $D_b$, the length scale of weathering decay, $\lambda$, and the landslide thickness, $T$. In a sense, Eq. (2) supports previous qualitative statements on the importance of rock type (Attal and Lavé, 2006; Roda-Boluda et al., 2018), which may physically relate to rock strength and regolith block size. Additionally, Eq. (2) combines the concept of physical weathering with the process of fragmentation, controlled by drop height, the latter of which less often considered in the geomorphological community. Future studies should also clarify whether $D_b$ is controlled by $\sigma_c$, which would make the equation more non linear but reduce the number of parameters to constrain. More complex models of fragmentation have been used to predict landslide GSD (De Blasio and Crosta, 2014; Ruiz-Carulla and Corominas, 2020), and may be better suited to model the full GSD, but Eq. (2), provided it is further validated, opens various interesting perspectives. For example, it suggests that seismically triggered landslides, which occur more often near ridges than

rainfall-triggered landslides (see Meunier et al., 2008; Rault et al., 2019) and thus are expected to have higher $H$, are more likely to deliver finer grains to the river systems, assuming they have a similar size and depth distributions (e.g., Marc et al., 2019). More generally it highlights the need for an investigation on how geomorphic factors (e.g., hillslope height, steepness, shape) modulate landslide runout, drop height, and connectivity to channels. Comparing hillslopes in various landscapes could

be easily attempted based on comprehensive landslide inventories (Tanyaş et al., 2017; Marc et al., 2018). Eq. (2) would also be well suited for landscape scale modeling of the input of various grain size into rivers (e.g., Benda and Dunne, 1997; Carretier et al., 2016; Neely and DiBiase, 2020), and thus, to better couple landslide and river dynamics in landscape evolution models (Campforts et al., 2020; Egholm et al., 2013).

## 4.2 Controls on the internal variability of the GSD and implications for future sampling

We found three sources of internal variability of landslide GSD: one associated with the lithology of the individual grains, as reported for Himalayan landslides by Attal and Lavé (2006), and two related to the location of the grains on or in the deposit, as reported for various rock avalanches (Crosta et al., 2007; Ruiz-Carulla et al., 2015). We discuss these observations first in terms of implications for bias and sampling procedure, and second in terms of physical process causing them.

The lithological difference is not likely to be a bias as long as the grains of different lithologies are randomly distributed in the
deposit: their sampling frequency should represent their relative abundance in the deposit. Spatial segregation on the surface of the deposit implies that to ensure a representative GSD, the sampling method should be performed ideally across most of the deposit, or at least over the different subunits of the deposit, before doing a weighted average with their relative area of contribution. Measurement based on sieving at a single site or local grain counts along a line or over a fraction of the landslide area may misrepresent the GSD and should be avoided. For large deposits where access is difficult, the use of pictures from a drone
may help with checking for segregation and potentially allow to reproduce the grid-by-number counting method using image analysis (e.g., Casagli et al., 2003; Attal and Lavé, 2006). However, this requires scaling each drone picture, and thus to deploy reference objects across the deposit which is not always practical, not counting the fact that such sampling will be unable to resolve fine grains ($< 30 - 100$ mm). In contrast, in the presence of a vertical segregation, where superficial and inner GSDs differ, it may be very difficult to estimate a GSD that is representative for the whole deposit. When possible, targeting the banks
of incised gullies may offer a good opportunity to characterize the subsurface of the deposit. Some applications mainly require the subsurface GSD, for example modeling the weathering of freshly fragmented bedrock in the landslide deposit and how they can contribute to solute fluxes (Emberson et al., 2016a, b). In contrast, the surface grains matter for sediment transport, and armoring may limit the mobilization of deeper finer grains. Additionally, in the case of a carapace, the question of how to combine the two end-member distributions would require an estimate of the relative thickness of the two end-member GSD,
which may be challenging. In the case of a less extreme segregation, as observed for LS-8 and probably LS-5, the proportion of coarse grains ($> 200$ mm) was similar on the surface and inside the deposit, and only the medium and especially fine grains were more abundant inside the deposit.

The process of kinetic sieving (Savage and Lun, 1988; Gray, 2018) is expected to cause vertical segregation (i.e., a coarser surface and finer subsurface) in granular flows, and a downslope segregation when shear is present, leading to boulder fronts as for LS-3. However, it should be noted that segregation is favored by transport along moderate slope gradients and tends to disappear for very steep chutes (Vallance and Savage, 2000). Although our gradient estimates are very rough, segregation mostly occurred for landslides with large transport distance, estimated as $\sqrt{L^2 + H^2}$, and least steep slopes (Table 1, Fig. S10). This excludes LS-12 with likely a complex displacement, and LS-16 for which the weak and clay-rich lithology, prone to form agglomerates, may not behave like a typical granular material. Still, it seems hard to explain with kinetic sieving why LS-9n was so clearly segregated downslope, in spite of its very modest size and displacement. Instead, we could hypothesize that on some landslide deposits, episodic reactivation of the scar and channel chute may have sprayed the deposit with finer debris, depositing preferentially near the apex of the deposit. Such mechanism might have happened on most of the landslides we have sampled (given their ages), but for now we cannot constrain its relevance without repeated monitoring of the deposits, which is left for future studies. Alternatively, for old deposits it is likely that fine materials could have been washed away by repeated storm events. This progressive washing of the fine grains would be consistent with the fact that the superficial deposits are very poor in fine materials, but have a proportion of coarse blocks fairly similar to the internal part of the deposit (for LS-5 and LS-8, Fig. 3). In these two cases, kinetic sieving may have been limited (although likely present in LS-8 to explain some downslope coarsening) and fines may have been preferentially washed out. On various parts of the deposit of LS-11 we did find finer materials when scraping off the top layer of gravels, consistent with this hypothesis. If such a process is expected to happen on all landslide deposits, superficial measurement of very fresh landslide may represent the bulk of the material (as perhaps LS-15 and LS-3, most recently failed and with high proportion of fine grains), and older deposit may require some correction as medium to fine grains may be underrepresented.

To conclude this discussion, it seems clear that several physical processes can add complexity to landslide deposit GSDs, and that deconvolving them and applying a process-based correction is not straightforward. More datasets are needed to better understand these sources of variability of the GSDs, for example with a more systematic sampling of very fresh landslides where fines should not have been washed out. Thus, we encourage such issues to be anticipated in future studies, and field work to be performed in a way allowing the spatial variability to be recorded. This would also enable future studies to include various landslide GSDs based on different assumptions or corrections. In this sense, collecting more measurements of landslides with both internal and superficial GSDs seems essential, especially when comparing young landslides with similar characteristics (lithology, height drop).

### 4.3 Implications for sediment transport in Taiwan

The landslide GSDs we report contain mainly gravel, but also a substantial fraction of boulders, which suggests that, after reaching floodplains and channels, the transport and evacuation of the material will require large floods. To compare these GSDs to typical shear stresses occurring in Taiwanese rivers, we use the shear stress map derived by Yanites et al. (2010b) from detailed measurement of the width, discharge and slope along the Peikang river. For a 10-year return flood with a discharge of 1000 $m^3.s^{-1}$, they found that shear stress, $\tau$, mostly ranged from about 60 to about 380 Pa. For mountain channels with

gradient typically about 2 % (Yanites et al., 2010b), these shear stresses correspond to flood heights between 0.3 and 2 m. To assess a threshold for bedload transport we computed the grain size $D$ for which the Shields number $\tau/(\rho/\rho_f - 1)gD$, with $\rho$ and $\rho_f$ the grain and fluid density, respectively, was above a transport threshold of 0.045 (Lamb et al., 2008). Similarly, for suspended load transport, we assessed for which $D$ the shear velocity, estimated as $U_* = \sqrt{\tau/\rho_f}$, was larger than the settling velocity of the grain $U_s$ as defined and calibrated by Ferguson and Church (2004) (Fig. 2, 7). Even for an above average 10-year return flood, less than 25% of most landslide deposits could be transported in suspension, except LS-10 and LS-16 with a suspended fraction of up to 50-70%. When accounting for bedload transport, the largest shear stress of $\sim 380$ Pa could not transport 5-25% of the deposits for about half of the landslides, especially LS-6, LS-9n and LS-13. Considering smaller, but not uncommon, shear stresses (60-140 Pa) would result in an immobile fraction of 20 to 40% for most landslides, and up to 80% for the three coarsest deposits.

Before discussing the implications of this, we highlight three main limitations which should be addressed by future work aiming at constraining the export of landslides deposits. First, the shear stress could not be adjusted to the local channel conditions in which the landslide occurred, neglecting specific width, discharge and gradient, as well as relations between gradient and critical Shields value (e.g., Lamb et al., 2008) or the influence of landsliding on the channel itself (e.g., Kuo and Brierley, 2014). Second, we ignored armoring effects, in which a superficial layer of coarse grains inhibits the mobility of finer grains (Parker and Sutherland, 1990). In our case, considering armoring could particularly reduce transport for deposits where coarse grains are segregated at the toe or surface of the deposit, such as for LS-2, LS-3 or LS-8. Third, we did not consider debris-flows and hyper-concentrated flows, which are frequent in Taiwan and sometimes reach the ocean (Dadson et al., 2005; Lin et al., 2005; Hsu et al., 2010), and which would enhance sediment transport given their higher fluid density. Despite these sources of uncertainty, our results suggest that in the relatively strong metasedimentary units, rapid evacuation of the sediment by suspension affects at most 30% of most of the deposits, and most of the transport occurs as bedload. Further, only the largest (10-year return or more) floods will transport substantial parts of the deposit, meaning that large landslide events may load channels with a pulse of coarse sediments requiring several decades to be evacuated. This is much longer than the transient pulse of enhanced landsliding (Marc et al., 2015) and suspended sediment transport (Hovius et al., 2011) observed after the Chi-Chi earthquake, which lasted less than 10 years, but is consistent with the $\sim 50$ yr timescales for enhanced lake sediment deposition (including bedload) after earthquakes observed in New Zealand(Howarth et al., 2012; Wang et al., 2020). This multi-decadal timescale for sediment export seems consistent with the very large alluviation of river channels in southern Taiwan, after intense flooding and landsliding triggered by Typhoon Morakot (Yanites et al., 2018), and which was still visible in 2015 (e.g., Taimali river), and at the time of writing in satellite imagery. Substantial aggradation, suspected to be long-term, was also observed after the Chi-Chi earthquake (Yanites et al., 2010a; Chen, 2009). More detailed modeling of the evacuation of landslide sediment (e.g., Yanites et al., 2010a; Croissant et al., 2017) could be combined with scenarios based on the detailed GSDs reported in this study to better quantify the dynamics and timescales of coarse sediment export after large landslides events.

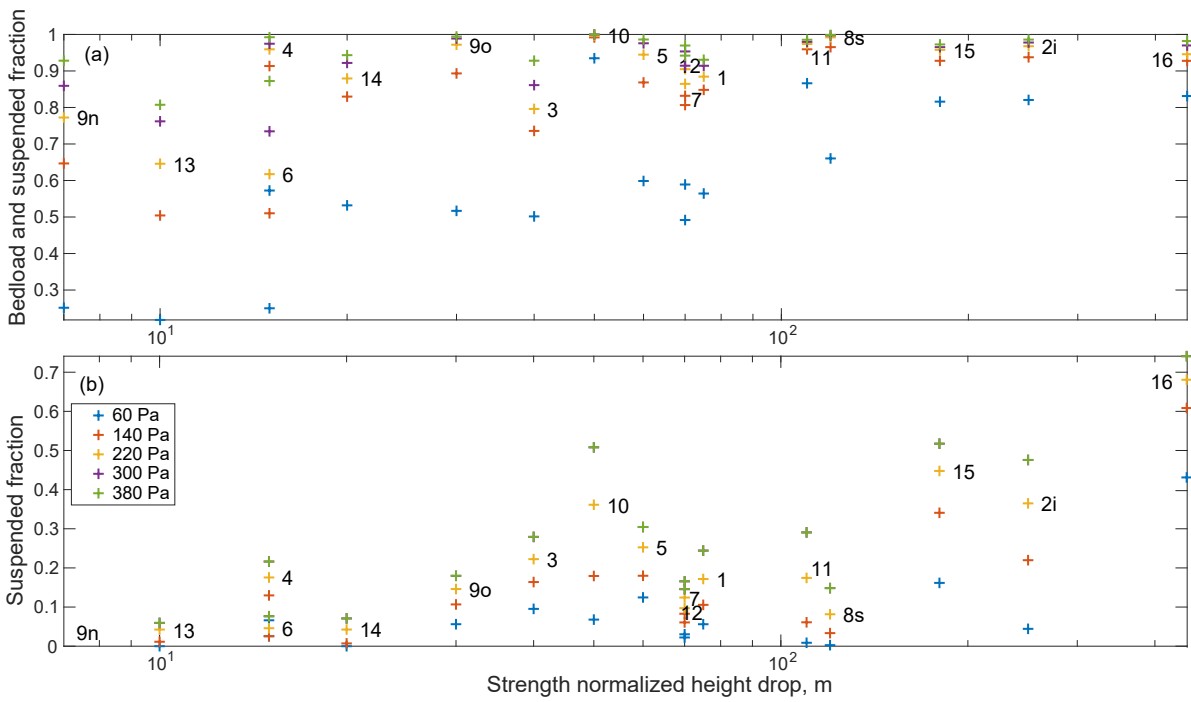

**Figure 7.** Fraction of the landslide GSD that could be transported as both bedload and suspended load (a) and only suspended load (b) as a function of the strength normalized landslide drop height (as in Fig. 4) and river shear stress during a 10-year return flood (illustrated by different colors). Note that in (b) suspended fraction at 300 and 380 Pa are identical.

## 5 Conclusions

We presented grain size distributions obtained from 17 landslide deposits in Taiwan. They have $D_{50}$ and $D_{84}$ consistent with landslides reported in previous literature, between 15 and 200 mm and between 60 and 600 mm respectively. We found that many deposits had significant spatial segregation in the downslope direction, with the lowest part of the deposits having 2 to

5  10 times coarser GSDs than the upper part of the deposits. For the three landslides in which we could sample the inner part of deposits, we also found GSDs were 3 to 10 times finer than their surface counterparts. The presence and intensity of this segregation cannot be attributed to a single process, but kinetic sieving and deposit reworking are likely to play important roles. This internal variability could bias results obtained from local sampling of GSDs, such as sieve samples from a single pit. Investigating the controls on landslide GSD variability, we observed a strong anticorrelation between the landslide drop

10  height, width and inferred scar depth and the GSD percentiles for all the landslides. Finer GSDs in the two landslides in non-metamorphosed, young sedimentary rocks can be well explained by normalizing the drop height by the rock strength. Further, modeling the source material median grain size with an exponential fining towards the surface, consistent with physical weathering models, we found that the reduction ratio from source material to landslide deposits matches the scaling proposed by Locat et al. (2006) and calibrated for rock avalanches with much larger volume and much higher point-load strength than the

ones we studied. Although future measurements on source rocks are needed for a complete demonstration, especially in terms of bedrock strength and fracture spacing, we suggest that simple geomorphic models coupling this fragmentation scaling with a model for regolith grain size (see Eq. (2)) could provide a physically-based first order model for the GSD input to rivers by landslides in active orogens. Such an approach could be implemented into landscape evolution models accounting for sediment transport. Indeed, from our deposits we also noted that even a 10-year flood may not be able to transport the coarsest fraction of many deposits, suggesting that floodplains and channels will likely need several decades to recover after large landslide events.

*Data availability.* The 28 GSD (for each landslide sub samples) are available in the Hydroshare open repository, together a shapefile with landslide locations and polygons derived from Google Earth. Marc, O., J. Turowski, P. Meunier (2021). Grain Size Distribution of 17 Taiwanese landslide deposits, HydroShare, http://www.hydroshare.org/resource/ade683be61e54fa5b60da97418a5f3df

*Author contributions.* OM designed the study and the field mission, performed all analyses and wrote the manuscript. JMT and PM provided input for the field methodology and the result interpretations, and edited the manuscript. All authors participated in collecting the grain size data in the field.

*Competing interests.* The authors declare no competing interests

*Acknowledgements.* The grain counting during the 2015 field campaign could not have been possible without the additional participation of Antonius Golly, Arnaud Burtin, Anne Schöpa and Niels Hovius, and they are warmly thanked for this. We also thank Kristen Cook and Anne Schöpa for contributing pictures, and Sebastien Carretier for pointing us to the literature on physical weathering models. We thank Mikael Attal, one anonymous reviewer and the AE Rebecca Hodge for their constructive reviews which helped clarify and improve this paper. We also acknowledge Camille Noûs, https://www.cogitamus.fr/camilleen.html , who embodies the broader scientific community and the collaborative and open nature of the creation and dissemination of knowledge, for its contribution to the design, methodology and interpretation of this study.

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
