# Peer review of "Controls on the grain size distribution of landslides in Taiwan: the influence of drop height, scar depth and bedrock strength."

_Earth Surface Dynamics, 2021_

## Author Response (AR1)

**Dear Editors,**

**Please find below a detailed response and list of edits made as a result of the comments and suggestions of both referees. We appreciate their constructive reviews and have almost systematically followed their recommendation leading to improved clarity and more in depth discussion.**

**We hope you will find the manuscript ready for publication in Esurf.**

**Sincerely,**

**Odin Marc, Jens Turowski and Patrick Meunier**

**Referee 2:**

Dear Editor, dear authors,

This paper presents a very valuable data set of grain size distributions supplied by landslides, and a simple yet seemingly effective model for predicted median grain sizes. This work will certainly attract the ESurf readership and is very worthy of publication.

However, the paper still needs Minor Revisions before it is ready for publication. The main issue I find is that is poorly written: some parts are very unclear, paragraphs are very long, some key points are buried in between other information, and there are a lot of grammatical errors. This makes the paper tedious to follow, and it's a shame because the data and the results are very exciting.

Additionally, a major weakness is that the authors make certain claims about their data without showing the evidence in graphs in the manuscript or the supplement.

Table 1, presenting the "raw" data, also needs more information and clarity in order to make this work fully reproducible, and more likely to be cited in future compilations.

Their claims on bedrock strength need to be a bit toned down given the lack of direct measurements and the large uncertainties in the published ones. Below, I list some of these issues in a page-by-page, line-by-line basis (in capitals, my suggestions for rewording).

Looking forward to see the revised version of the manuscript!

>> We understand the general comments of the reviewer (overall matching the other referee comments) and have done our best to address them, following the constructive and helpful suggestions. We have rephrased and edited the text, and added clarifications through out. A number of supplementary figures have been produced and added to the supplementary materials. Table 1 has been updated and extended following the referee advice, and lastly we have rephrased our presentation and discussion about rock strength.

Line 1: river --> riverS

>>OK

Line 2: "models have been developed for the grain size…" à models have been developed for ESTIMATING the grain size…

>>OK

Line 4: "until now relatively…" --> Until now, relatively…

>>OK

Line 5: distributionS, in plural, and we compare THEM

>>OK

Line 6: saying only "depth" is a bit unclear, as you could be referring to the scar/excavation depth, or the deposit depth, it would be better to make this explicitly clear

>> we now say "scar depth"

Line 8: "weaker from" --> weaker THAN

>>OK

Line 9: the word "deposit" is mentioned 4 times in this sentence, please rephrase (i.e. the first one could be change to "landslides", you could also say downslope vs. apex sectors)

>> Thanks for your suggestion, we replaced by landslide and  downslope vs. apex sectors.

Line 10: it's a bit unclear what do you men by "inside the landslides" – in pits, in incised sectors…? Also, "… inside the landslides that presented percentiles 3 to 10 times…" is not correct (it sounds as if you were selecting specifically landslides with those percentiles). Please change to "… inside the landslides (deposits?), which presented…"

>> We now specify "inside incised sectors of the landslide deposits, which presented ..."

Line 15: you do not have measurements of the original bedrock block size, so this sentence may sound a bit misleading. I suggested saying "estimated original bedrock block size" instead.

>>Ok

Line 16: D50 proportional TO the potential energy

>>OK

Line 17: "much stronger rock avalanches" is not correct, you mean avalanches that occurred in much stronger bedrock (the avalanches themselves are not "stronger").

>> Correct we rephrase to "was calibrated on rock avalanches with larger volume and stronger bedrock than those featured"

Line 18: for future modelING

>>OK

Line 19: riverS, in plural. Also, "aiming" may be better than "trying"?

>>OK

Line 2: parameter in singular. "Issues" sounds too informal, I suggest you change it to "processes". HazardS in plural.

>>OK

Line 5: "sedimentary structure" is an odd phrasing, why not say "stratigraphy"?

>>ok we now say : "which is ultimately archived in the stratigraphy"

Line 7: add comma after recent studies. Weathering on the critical zone reduceS, in present tense, it's not only a processes that occurred in the past.

>>OK

Line 10: "incompletely weathered" sounds odd (as if the expectation would be to be completely weathered), I suggest changing this to "only partially weathered".

>>OK

Line 20: you should write either "a physical scaling" or "physical scalingS", "derive physical scaling" is not grammatically correct.

>>OK

Line 27: how was the point-load strength of the bedrock measured? Also, state explicitly what H stands for (I assume it's drop height?)

>> We added in this sentence : "(with H the drop height of the center of mass, rho the rock density and g the gravitational acceleration),

" and "measured with a point-load test performed on rock sample from the sites."

Line 29: either add commas after the k2 and k1 values, or add a "being" (e.g. "k2=0.5, an empirical…" or "k2=0.5 being an empirical…").

>>Rephrased as : "Where $k_2=0.5$ is an empirical threshold for fragmentation, and $k_1=83.3$ is an empirical coefficient"

Line 30: landslideS, in plural

>>OK

Line 31: the deposit should increase, in present tense, not increaseD – you are describing a process that still occurs in present time.

>>OK

Line 32: Start a new paragraph with the sentence "However, subsequent studies…"

>>OK

Line 7: their model and resultS

>>OK

Line 10: change common to frequent?

>>OK

Line 11: add comma after studies: "based on these studies, we…"

>>OK

Lines 11-12: "intermediate size and depth landslides" sounds a bit odd, "landslides of intermediate size and depth" may be better.

>>OK

Line 12: typo on D50

>>OK

Line 13: "second, WE HYPOTEHSIZE that materials…"

>>OK

Line 14: "due to a reduction WITH DEPTH of the fracture density…"

>>OK

Line 17: riverS

>>OK

Line 20: "…where REPORTS of river GSDs exist in the literature"

>>OK

Line 27: "we DETAIL below", in present tense

>>OK

Lines 27-28: you say "for each deposit" twice in a row – why not write "how we constrained landslide characteristics and measured GSDs for each deposit"?

>>OK

Line 31: Specify in parenthesis their sizes and lengths range, and also which landslides do you consider "small" (e.g. < 0.1 km^2?)

>>Ok  we now specify that the "small" landslides were <1000 m2 , and were not very clearly visible in imagery.

Line 1: both sideS. In this line you talk 13 landslides being on the same lithological unit, which then on line 11 you mention again and explain is the Lushan formation. I suggest re-structuring this paragraph to make sure the information is given to the readers in coherent blocks and to avoid repetition.

Lines 3-4: the sentence before hints that you are going to explain the lithological unit of the other 4 landslides, but then you talk in this sentence about their geographic position, which for readers not familiar with Taiwan, provides no lithological information

>> To answer both comments we have restructured the paragraph to describe systematically the lithological units. We do not mention here anymore the formation that are only indicative for readers knowledgeable of Taiwan formations. They are stil given in the figure and table 1. The paragraph was rephrased to:

"To assess variability in GSD independent of rock type, 13 out of 17 landslides were chosen in the same geographic area, on both sides of the southern section of Taiwan Central Range, in relatively homogeneous lithological units composed of slate and slightly metamorphosed sandstone (Fig 1, Table 1).
LS-1 and LS-15 also occurred on moderately metamorphosed units, on both sides of the northern part of the Central Range, in black schist and in metasandstone intercalated with slate, respectively.
The two remaining landslides both occurred in unmetamorphosed units, made of sandstone and shale alternance for LS-10 which occurred in the emergent topography of Taiwan southern tip, and in shales of the Northwestern foothills for LS-16."

Lines 1-27: this paragraph is way too long, consider dividing in 2 or 3.

>> We have broken up the paragraph in 3 parts, about landslide type and age, landslide geographic and lithologic unit and last rock strength estimates.

Line 5: landslideS

>>OK

Line 8: yearS

>>OK

Line 13: change "last" for "other" or "remaining"

>>OK

Line 16: compared to the other UNITS, not landsdlides? Since you are talking about the bedrock lithology

>>OK

Line 20: "shales are about equally represented than sandstone" to "both shales and sandstone are equally represented"

>>OK

Lines 16-27: if I interpret this correctly, there are no available rock strength estimates for the Lushan formation (where most of your sampled landslides occurred), only for the Nanchuang. As the authors are surely aware, it is extremely tricky to convert point estimates of bedrock strength based on rock samples or Schmidt hammer measurements into meaningful, hillslope or outcrop-scale bedrock strength estimates, the uncertainties are often too large to provide meaningful information (that's why many other studies refrain from doing so). Even if these strength estimates were considered representative, there is a wide range of values within the same formation. Yet, towards the end of the paragraph, the authors say "… we expect the rest of the landslide from metasedimentary units to be stronger than around 100 MPa" (why? This sounds very unjustified at the moment), and finish the paragraph by providing quite specific point load strength values to each lithological unit. The limitations of these estimates should be explicitly stated, and the final estimates provided should be given as a wide range that truly reflects the very large uncertainties.

>> We agree to the reviewers, that this presentation is probably too bold given the variability. In the end the absolute value of strength (compressive and point load) is only useful to assess whether or not the Taiwanese landslide match the prediction of Locat , with the same coefficients…

To test the scaling, we only need relative strength, so what matters is that the strength of LS 10 and LS 16 are about three and ten times less than the other landslides strength, respectively.

So we now show in Fig 4 and 5 an error bar on these two landslides to account for the fact that strength could be 2 to 4 times less for LS 10 and 7 to 13 times less for LS-10 … What is important is that we assume that the landslides in the Lushan formation (as well as LS 1) have strength similar to LS 15. This is reasonable given they are both made of slate and metasediments, and is not disproved by the data in Fig 4 where all these landslides are similar.  So we have removed the direct strength estimates and now state :

"These measurements clearly make the case for highly variable rock strength and are far from encompassing the potential diversity of rock-type sampled by the studied landslides. Our goal here is not primarily to constrain the rock strength of individual landslides but to estimate the relative strength of diverse units. Based on the measurements reported above we make two assumptions. First, that the shales and sediments hosting LS-16 and LS-10 may be 7 to 13 times and 2 to 4 times weaker, respectively, than the metasediments hosting LS-15. Second, that the slates and metasandstones in the Lushan formation have similar strength than the ones in which LS-15, as well as LS-1 occurred and thus that these landslides can be compared without normalizing for strength."

We then refer to point load strength only during the discussion when these values are needed to compare to the prediction of Locat, and took care to specify :

"To compare with the prediction of \citet{locat_fragmentation_2006}, we have to assume a value for $D_b$ and use $\sigma_c$ values of 0.5 MPa for LS-16, 1.5 MPa for LS-10 and 5 MPa for the other landslides in metasediments. These point load are consistent with the typical unconfined compressive strength \citep{lin_effects_2008} after a dividing them by 20 \citep{chau_uniaxial_1996}."

"to which uncertainty on rock strength (i.e., probably a factor of two) should also be added."

Line 19: I assume that these data comes from Lin et al. (2008)? This should be appropriately cited.

>> We added the Lin et al 2008 citation twice.

Line 2: why estimating the scar slope rather than the pre-failure slope from previous imagery, or the hillslope slope from neighbouring undisturbed areas?

>> The slope was estimated based on the elevation model which is the SRTM so it does represent pre failure geometry. We now specify it, and have added :

"mostly 30m SRTM, predating all the studied landslides" and "An estimate of pre-failure scar gradient could be derived"

Line 3: "… the upper, respectively lower, volume…". It is very unclear what is meant by this. Do you mean you calculate minimum and maximum volume estimates?

>> Yes rephrased to : "We estimated a maximum and minimum landslide volume ..."

Line 4: for a few landslides you have field volume estimates (Table 1), so it would be useful to see how well do the V-A scaling and field estimates compare, at least in the supplementary material.

>> Yes it does match fairly well and we now added a figure in the supplement. That is why we state on L14 : "Nevertheless, these field estimates mostly fall within the bracket of the volumes estimated from global scaling relationships **(Fig Suppl 1),** lending some support to this approach "

We also cite this figure in the next sentence where we compare field volume and the regolith and bedrock scaling.

[Figure]

Line 7-9: I assume W in this equation means width, but you should tell the reader. Also, why if Google Earth imagery was good enough to map the landslides, it was not good enough to estimate scar areas? Was the contrast between scar and deposit not clear? Please explain the reader why the need for a scaling. It is also unclear if the landslides for which observed dimensions did not match the scaling were the only ones for which you had field dimension estimates or not. If they were the only landslides for which you had dimension observations, and hence you could use them to test if the scaling works or not, why do you decide to apply the scaling to the other landslides despite it not working for these ones? If you have dimension observations that match the scaling, then: (a) show the agreement in the supplementary material, and (b) explain why do you think the scaling fails for "a few landslides", and which ones. At the end of this sentence, Table 1 is quoted, but if I go to Table 1, I can't tell which landslides have measured vs. scaling-estimated areas.

>> 1. We have written out at the start of the paragraph that Ws is scar width.

2. To explain the need for a scaling we now state: " Using this scaling is necessary because in most cases, field or satellite observations did not allow to determine the end of the scar area and the transition to runout or deposition areas. Although approximate, this scaling is still preferable to using total landslide area, as it removes bias in volume estimation associated with variable runout length (e.g., Marc et al., 2019)."

3. The statement about disagreement between scar and slide was due some previous errors in the reporting and estimate of width and area (leading to some cases like LS 4 where $As=1.5Ws^2$ was larger than A, which was not reasonable…. ). After double checking all values, we find As to be between ~ 30% (for long runout like LS 2 or LS 11) and ~80% of A (for slides with very small runout, like LS 4, LS9n or LS12) which is plausible, and all scar areas used in our analysis for computing V and T are based on the scar width.

So we simply removed this part of the sentence.

Lines 10-15: it is not clear in Table 1 if to estimate the volume of each landslide, the V-A published relationship, or a fraction of a cone was used (are those the ones with "field volume" estimates? Then it should be made clear in this section).

>> Yes . We now say in L13 : "and yield only a first order "field volume" estimate (Table 1).

Lines 16-20: you are continuing your analyses choosing a "best estimate" fore each landslide, but which one you are choosing is not clear in Table 1. Please add a field in the table to make this clear, or mark in bold the numbers you consider as best estimate.

>> We added a best volume line and gather this new line and the best thickness with the lines about scaling to ease comparison.

Lines 23-24: this one sentence paragraph is a bit odd on its own, and it is also not clear to what correlations you mean (i.e. the ones in Figs. 4 and 5 only?), and what the method of "random boostrapped subsamples of the considerd samples" consists of and why do you think it is needed/suitable/desirable for your data analysis. Please clarify.

>> The other referee also question the need to apply this method. Given the data set is limited and there are no strong need to assess the impact of outliers, this approach does not bring much and is confusing. So, we simply removed these two sentences and present straightforward estimate or Pearson R or R2 (without uncertainties estimate).

Line 26: either you say "using the grid-by-number sampling METHOD" or "using grid-by-number sampling"

>>Ok, we use the second option now.

Line 29: it is unclear what advantaged had using a Phi scale and binning the data in the field measurements already (which as far as I'm aware, it is more useful for finer grain sizes, and the authors' measurements bin all grains finer than 2 mm anyway), please explain.

>>The half a phi scale yield a table which was printed and which could be filled with the number of measurement falling in each bins. This allowed to count faster and visualize pseudo histogram on the field compared to writing down a list of numeric value to be processed later…

No changes made.

Line 1: it would be useful to include how many grains have been sampled per deposit on Table 1, and to also mark there which landslides where segregated, and for which ones there are several measurements available (surface vs. sub-surface, toe vs. apex).

>> Yes, we have added 2 lines, one with the total grain count and one with the subsector counts.

Fig. 1: the background DEM or hillshade appers so dark that it is hard to see the topography – I suggest you adjust this so that it is easier to visualize

>> The DEM appears dark in the plain and western foothills which are much lower than the mountains in the central range. However, as elevation is relatively secondary relative to the rock type units, and we consider the pattern of valley as reasonably visible for the interested readers.

No changes made.

Line 5: "corser, respectively finer" – not sure what is meant by this. Do you mean that Locat's GSDs were 10 times coarser and Crosta's GSDs 10 times finer than your ones? Please rephrase to make this clear.

>> Yes. Ass suggested by the other referee we now write : "which were about 10 times coarser and finer, respectively, than all other studies."

Fig. 2: any ideas of why LS-16 is poorly fitted by log-normal and Weibull distributions? Was there anything different about this landslide?

>> Yes, a lot of the material is actually finer than the mm scale. So it should likely rather be described by a bimodal distribution with a fine mode (<1mm) and a coarse mode (around 10mm) as often observed in the landslides studied by Casagli et al, 2003. This is already stated in this section.

Fig. 3: in the caption, please add, for clarity: "Three examples of heterogeneity in grain size distributions ON A SINGLE LANDSLIDE DEPOSIT" or "on the same landslide deposit". Otherwise it's confusing as it may be interpreted as you are talking about heterogeneity across different landslides.

>> Ok we rephrased to: "Examples of three types of heterogeneity in grain size distributions within the same landslide deposit."

Line 9: was there anything special or different about those 4 segregated landslide deposits?

>> We discuss this at length in the Discussion section about segregation.

Lines 16-18: you should show figures/data supporting these claims, at least in the supplementary information.

>>OK we now show that in the supplement

Lines 22-24: this sentence is very unclear, it is very hard to follow the reasoning because it is very poorly written

>> To address both reviewer concern about this sentence we break it and expressed uncertainty (of a slightly different nature) separately for LS 5 and LS 2 . So we rephrased to:

"For LS-5 we considered the coarser distribution from the channel as more representative of the surface deposit. If the fan of LS-5 is representative of the deposit and if the channel is coarser because of some spatial segregation, the percentiles of its GSD would be about two to three times finer than the ones we have selected (Figure 3). In contrast for LS-2 we considered the internal GSD, because the superficial measurement recorded only what seems to be a carapace that over-represents coarse grains. Indeed, with about half of the distribution made of boulders ($>0.5\ m$) we consider that segregation to be exceptional within our dataset. Thus, in using LS-2i to study inter-event variability we assume that none of the other landslides had a carapace with similar strong sorting. Nevertheless, LS-2i percentiles may be biased towards finer grains, when compared to surface

deposits of the other landslides, and we assume the bias could be up to a factor of two to three, based on LS-5 and LS-8 (Fig. 3)."

Lines 27-28: I understand the logic behind the strength scaling, but this needs to be explained better. Also, as I mentioned in one of my comments above, I am a bit doubtful that the data you have on bedrock strength allows you to say that "strength is 3.3 times smaller" – please rephrase to acknowledge the uncertainty.

>> The other referee found this part was not clearly explained.  We have rephrased, being less accurate about strength, to:

"According to Eq (1), and assuming similar initial regolith material for all landslides, $D_{50}$ should decrease linearly with the ratio of drop-height, $H$, to bedrock strength, $\sigma_c$. Indeed, log-transfroming and fitting $D_{50}$ against $H$ for the 15 deposits from the metasedimentary units, which we expect to have relatively similar strength, we obtain $R^2=0.71$ and a power-law exponent of -0.64 (Figure 4). Including, LS-10 and LS-16 from weaker units yielded a substantially poorer fit (R2=0.31). Thus, to account for their weaker strength, we rescaled these two landslide drop heights by a factor of 3 and 10, respectively, at the center of values estimated from strength measurements (see methods). Then, we obtained correlation coefficient ($R^2 = 0.71$) and a best-fit power-law exponent of -0.78."

Fig. S4: I suggest that this figure is moved to the main manuscript; as this data shows important intrinsic correlations between landslide size/depth and drop height, which act in opposite directions in terms of their control on grain size. The correlations of D50 to other landslide metrics such as area, width, volume and depth should also be shown in the main manuscript, otherwise you are not showing the data supporting your claim of lines 30-31.

>> The data is also in Table 1, and it is fairly easy enough to read out that the widest landslide are also deepest and with the largest drop height. So I do not think we can say that we are not showing the data supporting this claim.

Line 30: when you presented the data, you mentioned the different formations, but now you talk about "metamorphosed units" – keep a consistent nomenclature throughout, it will make the paper easier to follow.

>> As answered in the previous comment, we now present the slides systematically with their lithological units and formations are only mentioned in Fig 1 and when comparing rock strength  estimates from the literature.

Lines 31-34: so if I understood correctly, you are arguing for a greater drop height, mechanical control on landslides GSDs vs. a more weathering/bedrock fracturing control. First, this needs to be written more clearly. Second, it would be interesting to see if drop height and landslide metrics correlate with hillslope length and slope; as that would allow making rough predictions of the GSDs that a hillslope may supply in the long-term even when no active landslides can be observed.

>> We are arguing for a combined control of depth and fragmentation, as detailed in the following discussion.

 We totally agree that understanding what geomorphic factor control drop-height seems essential. However we do not think focusing on 17 landslides could give a useful geomorphic answer. Rather what could be done is a statistical analysis on landslide runout and height drop, on various hillslopes and landscape. Such analysis is at hand based on large landslide inventories, and we now say it in the discussion, at the end of 4.1 :

"More generally it highlights the need for an investigation on how geomorphic factors (e.g., hillslope height, steepness, shape) modulate landslide runout, height drop, and connectivity to channels. Comparing hillslopes in various landscapes could be easily attempted based on comprehensive landslide inventories (Tanyas et al., 2017, Marc et al., 2018)."

Fig. 4: I find the colour scale very hard to see, perhaps because the points are too small. Consider making them bigger or using a colour scale with more contrast per increment. Also it would aid visualization if you join the red dots with their corresponding adjusted values with an arrow or a dashed line.

>> We have joined the original and rescaled drop height for LS 10 and LS16, also adding them an uncertainty bar for the rescaling. To enhance the color gradient we have saturated the color scale to 15 instead of 30, because only LS12 has a scar depth >15 m. We think this has improve greatly the clarity of the figure.

Lines 2-7: all these claims need to be backed up by showing these graphs, at least in the supplementary material.

>> We have added in the supplementary materials graphs showing the correlation or absence of correlations between GSD percentiles and Interquartile ratio… (Supplementary Figure 7 and 8 ).

Line 13: typo on D50

>>OK

Line 5: "proposed and validated by Locat et al., (2006) for large rock avalanches" should go between commas.

>>OK

Line 6: landslideS

>>OK

Line 17: where does this lambda=2 come from? What does it mean?

>> It means the weathering rate is divided by a factor e (2.7) every 2 meters. The value of ~2m was found in the Cohen et al 2010 study.

Line 19: how do you infer and quantify weathering intensity?

>> Intensity was a vague term that we have replaced by "weathering rate" yielding :

 "Here we consider that the **physical weathering rate (i.e., the rate of particle breakdown)** can be modelled with an exponential decay from the surface, with a characteristic length scale of $\lambda =2\ m$, consistent with previous modeling works \citep{cohen_marm3d_2010,anderson_rock_2013}.
Assuming the regolith median grain size at a depth z, D_r(z), to be proportional to the integral of the **weathering rate,** we modeled it as ..."

The cited literature (Cohen et al., 2010, Anderson et al 2013) describe in details how weathering rate can be modelled by incremental particle breakdown along depth profile, leading to profile of grain size alon gdepth.

Line 21: does this Db=1900 mm make sense based on observations of the study area (i.e. do you see fracturing patterns in the fresh bedrock of landslide scars, or are the biggest boulders in channels of these size?)? Would be nice to add a comment on that.

>>Good point, however, it is not very easy to relate this metric to punctual field observation (because it is a D_50 so it implies that the bedrock there are much larger and much smaller block size… ). Further it is the unweathered block size so the one below the soil, which typically should be sampled on cliff or landslide scars. As we also indicate in the text, given we match surface D50 and that inner D50 may actually be finer, the actual value of Db may rather be 600-1000 mm. And this value is in between previous measurements, and similar to grain size in the carapace of LS-2 which can be seen as closer to the original bedrock block size, as we now state :

"These latter values are intermediate between fracture measurement on surface outcrop in the US \citep{neely_bedrock_2019, verdian_sediment_2020} ranging from 10 to 400 mm, and the bedrock block size measured by \citet{locat_fragmentation_2006} on the scar of large rock avalanche, ranging from 600 to more than 10,000 mm. $D_b \sim 600-1000\ mm$ also matches quite well the range of $D_{50}$ to $D_{75}$ (450 to 900 mm) found in the carapace of LS-2 which may be a fair first order estimate of the original bedrock block size \citep{crosta_fragmentation_2007}."

Line 25: "…factor of two from THE PREDICTIONS OF Eq. (1)…"

>>Ok

Lines 5-6: what are the letters in Eq. (2) that represent the lenth scale of weathering and the landslide thickness? Please list them explicitly as you do for Db.

>> Lambda and T, now defined in the sentence.

Lines 6-7: Indeed some of these studies show, or imply, that there is a positive correlation between your Db and bedrock strength (σc). If Db is predicted by some function of σc, that would imply that landslides' D50 increases non-linearly with σc^2. I would emphasize more this important, potentially non-linear control of bedrock strength on landslide grain size.

>> Good point, we addedd the following sentence in the discussion:

"Future studies may also try to clarify whether $D_b$ is controlled by $\sigma_c$, which would make the equation more non linear but reduce the number of parameters to constrain."

Lines 16-17: mention that in cases of vertical segregation, incised gullies on the landslide deposit can be exploited.

>> Good point, we added : "When possible targeting the banks of incised gullies may offer a good opportunity to characterize the subsurface of the deposit."

Lines 25-27: this first sentence feels like it belongs in the Intro, or at least it should have also been mentioned there.

>> It is true this sentence is a state of the art sentence. However, the current structure of the introduction does not allow to fit easily this sentence. The process of segregation and its importance is already mentioned with the

reference to the Crosta et al 2007 works ( "often putting forward to explain their data the various mechanism of rock fragmentation and grain segregation (see Crosta et al.,2007, and references therein) " ) , but we feel, describing the different physical processes that can lead to segregation in the intro would be too long and distracting.

 Thus we prefer to leave the text as it is now.

Line 14: I suggest changing "complexify" for "add complexity", of much more common use

>>ok

Line 21: "… consistent with LANDSIDES REPORTED IN previous literature…."

>>Ok

Fig. 6: rather than showing these data by landslide number, it would be more useful to have landslide depth or drop height on the X axis (you can also add a small label of landslide number).

>> Good point, we have now remade the figure with the height drop (normalised by strength, ie as in Fig 4)

[Figure]

Answers to Referee 1 (M. Attal) :

Most comments of Referee 1 were about typos, grammatical errors or phrasing suggestions inserted in an annotated PDF. We have edited the manuscript accordingly and these changes are not reported here (but can be seen on the tracked changes PDF … )

Below we report only comments which were less straightforward to address, with our proposed edits.

P3 L4 : relations to local hillslope gradient ; Which I guess we related to erosion rates and time spent in the weathering engine, which links to statement on line 8 of previous page).
>> We now specify : "the local hillslope gradient, **as a control on the time spent in the weathering engine (Attal et al., 2015)."**

P4L5: About Disrupted landslides : May you be more specific? I had never come across that term, and a quick search did not lead to any significant insight. References?
>> We rephrased and now refer to the landslide classification:  "but most landslides could be called debris avalanches (Varnes, 1978), involving variable amount of regolith and bedrock, though LS-13 and LS-14 could also be called rock falls. LS-12, the largest event, may rather be a deeper rock slump, with moderate displacement, partly translational, partly rotational. "

P4L15 : Unclear - rephrase? I get the overall sense but it is awkward
>>We rephrased the sentence, focusing on the weakness rather than clay content. "In LS-16 many coarse rock fragments ($>10$ cm$) were crumbling when touched, highlighting the weakness of this rock compared to the other units."

P4 L21 : Be more specific (about "more irregular")
>> We replaced "more irregular" by "less frequent" and now rephrased to : "In contrast, the metasandstone are intercalated with less frequent slates, often stronger than the Nanchuang shales (Lin et al., 2008)"

P5 L3-6 : Double use of respectively is confusing. Rephrase.
>> As suggested by the other referee we now state :
"We estimated a maximum and minimum landslide volume ….  assuming the scar was mobilizing bedrock or soil, respectively"

P5 L 13-16 : Simplify. Takes a while to understand the sentence.
P5 L17 "Or the averaged the soil and the bedrock scaling" ????
P5  L19 -21 : Why? What is this based on? I think the argument in this section can simplified.

>> We have substantially rewritten these few sentence, also following the other referee advice.
There was a typo in the sentence L17 which did not help.  Overall we rephrased the sentence from L17 to 21. The point is that A-V scaling are crude way to estimate volume, and if we want a more accurate estimate we need to choose the soil or bedrock scaling...
When we have a volume from the field we do not need to use scaling. For the other slides we had to make choices based on their size. Pure bedrock for the large ones, intermediate for the other given most of them still seemed to involve a mixture of fresh bedrock and soil/regolith.
We now state :
"Still, for the deposit where we could not obtain a field estimate, better constrained volume estimate could be obtained by choosing one of the two scaling relationships. We note that the field volume estimate of LS-3 and LS-4 is similar to the estimate from the soil scaling relationship (Fig. S1). This is consistent with the observation that they were composed of rock debris with a yellowish color that indicates advanced weathering, and contained fresh vegetation debris (see Fig. 1). For LS-7 and LS-11, which were clearly involving mostly fresh bedrock, the field volume estimate better matches the bedrock scaling (Fig Suppl. 1). Thus, where field volumes were

lacking, we used the bedrock estimate for the largest landslides ($W_s > 50$) within which the rock looked mostly fresh (i.e., LS-1, 2, 8, 12). Some other landslides (LS-5, 6, 9o, 16), featured a mixture of soil and rock material. Consequently, we used the average of the soil and bedrock scaling to estimate their volume. The best estimate for each landslide was divided by its scar area to obtain an estimate of scar thickness (Table 1)."

P5 L23-24 : About Bootstrapping. Landslide samples? Why is this needed? I don't understand what this refers to.
>> The other referee also question the need to apply this method. Given the data set is limited and there are no specific need to remove outliers, this approach does not bring much and is confusing.
So we simply removed these two sentences and present straightforward estimate or Pearson R or R2 (without uncertainties estimate).

Table 1: Occurrence Year : decimals on the years looks a bit odd, but ok if you are trying to make the point they occurred at different seasons?
>> The referee is right. In the end the exact month does not really matter either so we simply remove the decimals and specified (Occurrence Year (Jul-Sep) ) which is the period where all dated landslide have occurred.

P6 L7: "by volume and by number" What do you mean ? Why do you think that is the case? Do you mean, compared to the surface estimates?
>> Yes by definition if surface GSD is different from inner GSD, if we remix the deposit it is likely that the inner volume is larger than would be a surface layer (where a carapace could form or where fine could have been washed away) and thus that the mixed deposit would be more representative of an inner GSD.
The "by volume and by number is unnecessary and confusing and we now simply state:
"while the latter likely represents a remixing from surface and internal parts, and thus must be closer to the inner GSD than what would be derived from surface measurements only."

P7 Fig 1 : > symbol is confusing: remove and say that all other landslides are in Lushan? Add the formation to which LS16 belongs?
>> We removed the ">", and as recommended stated all other LS are in Lushan, and added an open symbol for the Cholan/Chinshui Shale with LS-16

P8 : I count 20 distributions. Figure 1  caption says 16 landslides. In panel (a), it says n=17. Consistency?
> There was some confusion and we reported wrong numbers in some places.
There are 17 landslide, but three of them are separated in "surface" and "inner" GSD, which makes 20 GSD. Now we only refer to these correct values in the text and figure, and state in Fig 2 Caption : "cumulative distribution function for the 17 sampled landslide deposits. Note that we show surface and inner distributions for LS-5, LS-8 and LS-2, and thus 20 distributions in total."

Figure 2 : I don't understand: these are data, aren't they? Where is the best fit?
Why do you have two panels? The two panels seem to show two subsets: are these differentiated on purpose or just to facilitate display? If they are two different categories, why does a log-normal fit the former and a Weibull the latter? Is there a reason why some lines are dashed while others are not? Please give more information, thank you!

P8L5 : I think you need to give more details in the caption (of Fig 2), and some information about the goodness of the fits, if you want to talk about that? What does a distribution with a good fit to one of these two models look like? Why does that matter?
I guess if all distribution fit well a particular model, it offers good perspectives for modelling? But you need to explain that to the reader. The fact the fits are not so good precludes the use of a given distribution, which is a result!

>> These curves are only data and the fit were not shown. Indeed the two panel and the dashed line are simply for visibility. We made two group based on if the best-fitting distribution was Weibull or Lognormal but it is somewhat arbitrary.
To clarify we have added two supplementary figures (Fig S4 and S5) showing the CDF of each GSD with the best fit CDF for a Weibull and Log normal distribution. Allowing to see which distributions are best fit by a Weibull or a Log normal distribution. LS-16 is well fit by none (as it likely bimodal) and a few case are equally well fit by both distribution. We also conclude this paragraph with :

"These two subgroups imply that we cannot prescribe a given distribution form to model landslide GSDs, and future work may aim at understanding why some landslide GSDs obey one or the other distribution."

P10 L 18 :  Show these data ?
>> OK, we now show the correlation between D50 and other percentiles in Fig Suppl 7. We also show the Lack of correlation between the various percentiles and the Interquartile Ratio in Fig Suppl. 8.
This paragraph was updated and clarified :
"The percentiles of the GSDs are highly correlated with linear correlation coefficients R^2 >0.9 between $D_{50}$ and $D_{16}$, $D_{25}$, $D_{75}$, $D_{84}$ and $D_{90}$ (Fig. S7). We note that more scatter is present for D_16 and D_90, suggesting that $D_50$ is a good proxy for the bulk of the distribution but does not completely capture the variability in their tails.
The interquartile ratio (here $D_{75}/D_{25}$) which characterizes the span of grain size in the distribution, is also independent of the other percentiles (Fig. S8)."
"

P10 L21: "forming a carapace". How do you know this is not the case in all the other deposits?
>> We have no proof of this, but we see that LS-2 channel surface is extremely coarse almost half of the particles are larger than 0.5 m , whereas other landslides have typically less than 10 % of these large particles (so a few boulders at the surface only). In other site with incised sector the difference between surface and inner GSD was also far less extreme. So to be clear about this we now state:
"In contrast for LS-2 we considered the internal GSD, because the superficial measurement recorded only what seems to be a carapace that over-represents coarse grains. Indeed, with about half of the distribution made of boulders ($>0.5\ m$) we consider that segregation to be exceptional within our dataset. Thus, in using LS-2i to study inter-event variability we assume that none of the other landslides had a carapace with similar strong sorting."

P10 L22-24 : Long sentence difficult to understand
>>  To address both reviewer concern about this sentence we break it and expressed uncertainty (of a slightly different nature) separately for LS 5 and LS 2 . So we rephrased to:

"For LS-5 we considered the coarser distribution from the channel as more representative of the surface deposit. If the fan of LS-5 is representative of the deposit and if the channel is coarser because of some spatial segregation, the percentiles of its GSD would be about two to three times finer than the ones we have selected (Figure 3).  In contrast for LS-2 we considered the internal GSD, because the superficial measurement recorded only what seems to be a carapace that over-represents coarse grains. Indeed, with about half of the distribution made of boulders ($>0.5\ m$) we consider that segregation to be exceptional within our dataset. Thus, in using LS-2i to study inter-event variability we assume that none of the other landslides had a carapace with similar strong sorting. Nevertheless, LS-2i percentiles may be biased towards finer grains, when compared to surface

deposits of the other landslides, and we assume the bias could be up to a factor of two to three, based on LS-5 and LS-8 (Fig. 3).”

P10 L29: Add respectively. The approach seems to be the wrong way around: I think it would be better to try to fit all data together without massaging, demonstrate that the points that don't fit are in different lithology, and then propose an approach based on the mechanical properties of the rocks to normalise for strength (and show it works!)
>> The other referee found this part was not clearly explained.  We have rephrased to:
“According to Eq (1), and assuming similar initial regolith material for all landslides, $D_{50}$ should decrease linearly with the ratio of drop-height, $H$, to bedrock strength, $\sigma_c$.
Indeed, log-transfroming and fitting $D_{50}$ against $H$ for the 15 deposits from the metasedimentary units, which we expect to have relatively similar strength, we obtain $R^2=0.71$ and a power-law exponent of -0.64 (Figure 4). Including, LS-10 and LS-16 from weaker units yielded a substantially poorer fit (R2=0.31). Thus, to account for their weaker strength, we rescaled these two landslide drop heights by a factor of 3 and 10, respectively, at the center of values estimated from strength measurements (see methods). Then, we obtained correlation coefficient ($R^2 = 0.71$) and a best-fit power-law exponent of -0.78.”

P11 L 7 :  Please show all these data and correlations, even if only in SI!
 >> The last part of the sentence was erroneous, because we had not accounted for the fact that low percentiles of LS-16 are not meaningful (We just know that D16 and D25 are below 2 mm … ) Removing this landslide we have R2=0.2 which is not very exciting … So we prefer to simply say:
“Thus, more data is needed to understand the controls on the spread of the landslide GSDs.”

P12 L 21 and L 26 (about Db) : Where does this number (Db=1900mm) come from? It looks like a part of this sentence is missing.
 Explain that (Db=1900mm) in the previous paragraph: that seems like a sensible way of calibrating the model, and Fig 5a confirms that.

>> First we rephrased L21 to introduce the link between D_i and D_b without imposing yet the value of D_b.
“Averaging $D_r$ from 0 to $T$, the mean scar thickness, and choosing a value for $D_b$ we can obtain $D_i = D_b (1- \lambda/T (1-e^{-T/\lambda}) )$ for each landslide (Fig 5a).” (See the article version for correct formatting of the in-line Equation)

Then : Without field measurements of $D_b$, we note that assuming $D_b=1900\ mm$, the ratio of the modeled $D_i$ to the measured $D_{50}$ agrees with Eq (1) ...

And later we specified :  and thus field measurement of regolith and bedrock GSD may need to be compared to a model with $D_b \sim 600-1000\ mm$ to which uncertainty on rock strength (i.e., probably a factor of two) should also be added. These latter values are intermediate between fracture measurement on surface outcrop in the US \citep{neely_bedrock_2019, verdian_sediment_2020} ranging from 10 to 400 mm, and the bedrock block size measured by \citet{locat_fragmentation_2006} on the scar of large rock avalanche, ranging from 600 to more than 10,000 mm.”

P15 L2 : Awkward phrasing : “but is hard to constrain its relevance without repeated monitoring of the deposits.”
>> Was replaced by : but could not be constrained without repeated monitoring of the deposits.”

P15 L 3, 4 : The geometry of what ? Contorted long sentence to fragment.
>>We meant the geometry of the landslide path (distance from scar to deposit, slope angle).
Re-reading this part we feel it is quite speculative, and we may simply remove the sentence and end on this rephrased one:  “Such mechanism might have happened on most of the landslides we have sampled (given their

ages), but for now we cannot constrain its relevance without repeated monitoring of the deposits, which is left for future studies."

P15 L11 : "very fresh landslide may already represent the bulk of the material" Why Already ?
>> Already was confusing, and could be simply removed.

P15 L 29 : References? What is ro? ro_f?
>> The reference is Church and Ferguson 2004, for settling velocity, for the Shields number we added the reference of Meier-Peter and Muller, 1948, and Lamb et al 2008 as its Fig 1 show nicely 0.045 as a median value for slopes about 1%.
We added : "with Rho and Rho_f the grain and fluid density, respectively, ... "

P15 L 33: Sentence structure? What do these shear stresses typically represent?
>> We simplified the sentence to: "However, the lower range of estimated shear stresses (60-140 Pa) would result in an immobile fraction of 20 to 40\% for most landslides"

Then to better illustrate the meaning of the Shear stress we added just after defining the range of shear stress:
"For mountain channels with gradient typically about 2 %, these shear stresses correspond to flood heights between 0.3 and 2 m."

P16 L3 : "+ effect of slope on threshold for entrainment, e.g., Lamb et al. Also the adjustment of channels to landslides, see nice study by Kuo and Brierley: The influence of landscape connectivity and landslide dynamics upon channel adjustments and sediment flux in the Liwu Basin, Taiwan"
>> We rephrased to: "First, the shear stress was not adjusted to the local channel conditions in which the landslide occurred, neglecting specific width, discharge and gradient, as well as relations between gradient and critical shields (e.g., Lamb et al 2008) or the influence of landsliding on the channel itself (e.g., Kuo and Brierley, 2014)."

P16 L5: References asked.
>> This sentence, about armoring was more about expectation than about specific work. We have rewritten this sentence to make this clear and included a classic reference about armoring in the previous one :
"Second, we ignored armoring effects, in which a superficial layer of coarse grains inhibits the mobility of finer grains (Parker and Sutherland, 1990). In our case, considering armoring could particularly reduce transport for deposits where coarse grains are segregated at the toe or surface of the deposit, such as for LS-2, LS-3 or LS-8."

P16 L6 : Although once a debris flow dumps all its coarse sediment in a channel, it becomes very difficult for the river to mobilise the largest grains - only exceptional events will do.
Another point to consider: whether the landslide connects to the river network, e.g. Li, G., A. J. West, A. L. Densmore, D. E. Hammond, Z. Jin, F. Zhang, J. Wang, and R. G. Hilton (2016), Connectivity of earthquake-triggered landslides with the fluvial network: Implications for landslide sediment transport after the 2008 Wenchuan earthquake, J. Geophys. Res. Earth Surf., 121, 703–724, doi:10.1002/ 2015JF003718.

>> We agree with this comment in a general sense. However, in Taiwan hyper concentrated flow can directly reach the shoreline and thus drop sediments in marine canyon (as discussed in some references). We now added :
"debris-flows and hyper-concentrated flows, which are frequent in Taiwan and sometimes reach the Ocean"
In terms of the landslide connectivity, clearly if landslide are deposited on slope they won't be transported in a channel, as bedload but rather by other hillslope processes which are poorly constrained (sheet flow, rilling, landslide remobilization etc). However, even if this is important in terms of budget we consider it beyond the scope of this paragraph.

We now specify in the first sentence of this paragraph, that we limit this discussion to transport in channels and floodplains :

"The landslide GSD we report contain mainly gravel, but also a substantial fraction of boulders, which suggests that, **after reaching floodplains and channels,** the transport and evacuation of the material will require large floods. "

Fig 6 Caption : Why not using 400, as in the text?

>> Well, we sampled Tau by 80 Pa increment so it made sense to stop at 380 Pa, and 400 was just an approximate upper boundary for the maximum stress in Yanites et al 2010. So to avoid confusion we rather change the text and now say :

"tau, ranged from about 60 to about 380 Pa" .

---

## Editor Decision (ED2)

[revised manuscript text omitted]
         | SI/Sd                | Sl/Sd   | Sl/Sd         | SI/Sd   | SI/Sd           | SI/Sd                | Sl/Sd   | SI/Sd                | Sh/Sd         | Sl/Sd   | SI/Sd         | SI/Sd   | Sl/Sd   | Msd     | Sh      |
|                                                                                                                              |         |               |                      |         |               |         |                 |                      |         |                      |               |         |               |         |         |         |         |

of grains either looked fairly homogeneous, or because of lack of time their systematic counting within different rock types could not be done. To study the variability between various landslides we obtained an overall surface GSD by summing the grain counts from both lithology of LS-7 and from the different sectors of the deposits with spatial segregation. We did not use an area weighted sum (Ruiz-Carulla et al., 2015) because the upper, middle (when differentiated) and lower sections of the deposits represented roughly similar proportion of the surface of the deposits, and we obtained count variations below 10% from the different subsections (Table 1, Fig 3). When we measured both internal and superficial GSDs we had to select one of

them as representative for the comparison to other landslide properties (see section 3.2).

**3** Results**

5

**10 3.1 Landslide grain size distributions and their internal variability**

Before averaging spatial variability, the landslide GSDs have 50th and 84th percentiles ranging from about 15 to 200 mm and about 60 to 600 mm, respectively. This is consistent with the range of observations from previous studies, except the large rock avalanches from Locat et al. (2006) and the volcanic rock avalanches from Crosta et al. (2007), which were about 10 times coarser and finer, respectively, than all other studies. LS-2s and LS-16 are much coarser and finer than the rest of the studied

15 landslides, respectively. Interquartile ratios vary between 3 and 15, but we note that 13 out of 20 GSD have an interquartile ratio of 3 to 6, while only LS-1, 3, 5i, 5s, 8i, 15 and LS-16 have larger spreads (Fig. 2). All distributions seem unimodal, except LS-16 with more than 40% of the grains finer than 2 mm, likely containing a second, sub-millimetric mode that could not